# Lipid accumulation controls the balance between surface connection and scission of caveolae

**Madlen Hubert[1], Elin Larsson[1], Naga Venkata Gayathri Vegesna[1], Maria Ahnlund[2], Annika I Johansson[3], Lindon WK Moodie[4†], Richard Lundmark[1]\***

[1]Department of Integrative Medical Biology, Umeå University, Umeå, Sweden; [2]Swedish Metabolomics Centre, Department of Forest Genetics and Plant Physiology, Swedish University of Agricultural Sciences, Umeå, Sweden; [3]Swedish Metabolomics Centre, Department of Molecular Biology, Umeå University, Umeå, Sweden; [4]Department of Chemistry, Umeå University, Umeå, Sweden

**Abstract** Caveolae are bulb-shaped invaginations of the plasma membrane (PM) that undergo scission and fusion at the cell surface and are enriched in specific lipids. However, the influence of lipid composition on caveolae surface stability is not well described or understood. Accordingly, we inserted specific lipids into the cell PM via membrane fusion and studied their acute effects on caveolae dynamics. We demonstrate that sphingomyelin stabilizes caveolae to the cell surface, whereas cholesterol and glycosphingolipids drive caveolae scission from the PM. Although all three lipids accumulated specifically in caveolae, cholesterol and sphingomyelin were actively sequestered, whereas glycosphingolipids diffused freely. The ATPase EHD2 restricts lipid diffusion and counteracts lipid-induced scission. We propose that specific lipid accumulation in caveolae generates an intrinsically unstable domain prone to scission if not restrained by EHD2 at the caveolae neck. This work provides a mechanistic link between caveolae and their ability to sense the PM lipid composition.

**\*For correspondence:**
richard.lundmark@umu.se

**Present address:** [†]Department of Medicinal Chemistry, Uppsala University, Uppsala, Sweden

**Competing interests:** The authors declare that no competing interests exist.

## Introduction

Caveolae are bulb-shaped invaginations of the plasma membrane (PM), enriched in cholesterol (Chol), sphingolipids, and the integral membrane protein caveolin1 (Cav1) (*Parton and del Pozo, 2013*). Caveolae are present in most cell types, with a particularly high density in endothelial cells, adipocytes, and smooth muscle cells. Absence or malfunction of caveolae is associated with a number of conditions such as lipodystrophy, muscular dystrophy, and cardiovascular diseases (*Cohen et al., 2004*; *Pilch and Liu, 2011*). Although the mechanism of how caveolae dysregulation drives the phenotype of disease is not well understood, they have been proposed to serve as signaling platforms, endocytic carriers, and PM reservoirs involved in mechanoprotective processes or lipid buffering (*Sinha et al., 2011*; *Parton and del Pozo, 2013*). In adipocytes, which are key lipid homeostasis regulators, caveolae are estimated to account for more than 50% of the cell surface area (*Thorn et al., 2003*). The clinical manifestation of caveolae loss in both patients (*Cao et al., 2008*; *Kim et al., 2008*; *Hayashi et al., 2009*) and mouse models (*Razani et al., 2002*; *Liu et al., 2008*) reveals severe malfunction of adipocytes, in addition to other cell types involved in lipid turnover and storage.

Caveolae biogenesis is tightly coupled to the PM lipid composition and is thought to be driven by Chol-sensitive oligomerization of Cav1 and subsequent association with the cavin coat proteins (*Figure 1—figure supplement 1*; *Parton and del Pozo, 2013*). Cav1 is embedded into the lipid bilayer via its intramembrane and scaffolding domains, which interact with Chol (*Parton and del*

*Pozo, 2013*). Chol depletion from the PM causes caveolae disassembly, leading to the disassociation of cavins from Cav1, which then disseminates throughout the PM (*Rothberg et al., 1992*; *Morén et al., 2012*). Relative to the bulk PM, phosphatidylserine, glycosphingolipid (GSL), and sphingomyelin (SM) have also been proposed to be enriched in caveolae (*Ortegren et al., 2004*; *Singh et al., 2010*; *Hirama et al., 2017*). However, it has not been determined whether specific lipids are enriched in caveolae in living cells and, furthermore, how exactly they influence biogenesis, that is through a general physical effect on the membrane bilayer or via direct interactions with the caveolae coat proteins. Additionally, it is not known whether these lipids can diffuse freely in and out of the caveolae bulb or if they are sequestered by interactions with the caveolae coat and membrane curvature restraints.

Although caveolae are typically associated with the PM as bulb-shaped invaginations, they also exhibit dynamic behaviour including flattening (*Nassoy and Lamaze, 2012*), short-range cycles of fission and fusion with the PM, and endocytosis (*Pelkmans and Zerial, 2005*; *Boucrot et al., 2011*; *Morén et al., 2012*). Caveolae are stabilized at the cell surface by the ATPase Eps–15 homology domain-containing protein 2 (EHD2), which oligomerizes around the neck of caveolae in an ATP-dependent mechanistic cycle, restraining caveolae scission (*Figure 1—figure supplement 1*; *Morén et al., 2012*; *Stoeber et al., 2012*). EHD2 extensively colocalizes with Cav1 and most of the membrane-associated EHD2 is found in caveolae (*Morén et al., 2012*). Although not considered part of the caveolae coat, EHD2 appears to be a critical component for maintaining caveolae integrity in terms of surface attachment. All the aforementioned proposed functions of caveolae depend greatly on whether they are surface associated or released, but it is not clear how the balance between these states is controlled physiologically. Furthermore, the biological function of their atypical dynamics remains elusive (*Hubert et al., 2020*).

Lipids are also thought to influence caveolae dynamics, for example addition of bovine serum albumin-complexed lactosyl ceramide (LacCer) to human skin fibroblasts and elevated levels of Chol in adipocytes have been proposed to reduce the number of surface-connected caveolae and increase their mobility (*Sharma et al., 2004*; *Le Lay et al., 2006*). However, despite their essential structural role in caveolae, little is known about how they influence caveolae biogenesis and dynamics. This knowledge gap can be partly attributed to limitations associated with the current methods employed to study such phenomena. Drugs such as statins, that inhibit Chol synthesis, require multi-day treatments, and, in addition to altering transcriptional regulation, may also elicit major secondary effects (*Crescencio et al., 2009*). This results in downregulated expression of Cav1, making it difficult to decipher between the effects of Chol levels on caveolae biogenesis and caveolae dynamics. In addition, drugs such as myriocin, which inhibit sphingosine synthesis, will also affect the levels of all sphingolipid species, thus hampering direct conclusions. BSA-coupled Bodipy-LacCer has been used as a fluorescent marker of endocytosis (*Puri et al., 2001*; *Singh et al., 2003*; *Sharma et al., 2004*; *Singh et al., 2006*), but this procedure involved PM loading at 4°C followed by a temperature shift to 37°C, known to heavily influence membrane fluidity and exacerbate endocytosis (*Kleusch et al., 2012*). Previous studies have indicated that lipids might influence both caveolae numbers and their dynamics, but have been unable to address whether caveolae dynamics respond directly to alterations in PM lipid composition and if the proposed effects are dependent on concentration or species of different lipids present. In general, our understanding of the levels of quantitative changes in PM lipid composition that can be sensed and controlled is relatively sparse, not to mention the alteration in lipid composition required to influence caveolae. It is also not known if lipids could affect caveolae dynamics by changing the composition of the caveolae bulb or the surrounding membrane and whether the proposed effects on caveolae mobility are caused by direct effects on caveolae scission from the cell surface.

To address this, we aimed to rapidly and selectively manipulate cellular membrane lipid composition in a system where both the lipids and caveolae could be tracked. Here, we have applied fusogenic liposomes that allowed us to directly insert specific unlabeled or fluorescently labeled lipids into the PM of living cells and study their effect on caveolae dynamics. Our data show that a relatively small increase in glycosphingolipids and Chol results in their accumulation in caveolae, reducing the caveolae neck diameter, and driving caveolae scission from the PM. EHD2 was found to counterbalance the stability of caveolae in response to lipid composition and, in accordance with recent studies (*Morén et al., 2019*; *Matthaeus et al., 2020*), we describe a key regulatory role of EHD2 in lipid homeostasis.

## Results

### Lipids rapidly insert into the PM of living cells via liposome-mediated membrane fusion

As a tool to study the effects of altered lipid composition on caveolae dynamics, we employed fusogenic liposomes. This allowed rapid insertion of lipids into the PM of HeLa cells via membrane fusion (*Figure 1A*). To assess the effect of specific lipids, Bodipy-labeled analogues of sphingolipids (Cer, SM $C_5$ and SM $C_{12}$), GSLs [ganglioside GM1 and lactosyl ceramide (LacCer)], Chol and phosphatidyl ethanolamine (PE) (*Figure 1—figure supplement 2A*) were incorporated into liposomes [DOPE/Dotap/Bodipy-tagged lipid (47.5/47.5/5)] (*Csiszár et al., 2010*). Liposomes had diameters between 160 and 300 nm (*Figure 1—figure supplement 2B*) and an average fluorescence per liposome of 600 a.u (*Figure 1—figure supplement 2C*). The Bodipy fluorophore allowed us to track and quantify lipid incorporation in the PM and study their colocalization with caveolae components. To ensure the observed effects were not significantly influenced by the fluorophore motif, the results were verified with unlabeled lipids. Liposome fusion with the PM of HeLa cells occurred immediately upon contact and the lipids were rapidly distributed throughout the basal membrane, as observed using live cell total internal reflection fluorescence (TIRF) microscopy (*Figure 1B* and *Figure 1—figure supplement 2D*, exemplified by LacCer). The total fluorescence attributed to the Bodipy motif increased uniformly in various regions of interest (ROIs) (*Figure 1B*, line graph). Lipid incorporation was similar for all cells in the population and the even distribution of lipids in the PM was observed for all lipid species (see *Figures 1B, F*, *2C* and 4A–D, *Figure 2—figure supplement 2A*, *Figure 2—videos 1*, *2*, *3*, *4*, *Figure 3—figure supplement 1A*, *Figure 4—figure supplement 2A–D*). Occasionally, bright stable spots and some enrichment of lipids in cellular protrusions were observed independent of lipid species.

To determine the amounts of lipids that were incorporated in the membrane through liposome fusion, we used quantitative mass spectrometry on whole cells as 90% of these lipids are located in the PM (*Lorizate et al., 2013*). The method was verified by altering the lipid composition using myriocin (24 h treatment) or sphingomyelinase (SMase, 2 h treatment), which are known to lower the levels of sphingomyelin (*Gulshan et al., 2013*). Analysis showed that these treatments drastically decreased SM(d18:1/16:0) levels, the major endogenous species of SM (*Figure 1C*). Next, we incubated cells with fusogenic liposomes containing Bodipy-labeled LacCer or SM $C_{12}$, and analyzed the lipid composition by liquid chromatography electrospray ionization tandem mass spectrometry (LC-ESI-MS/MS). The detected levels of endogenous LacCer(d18:1/16:0) and SM(d18:1/16:0) in untreated control samples were in agreement with previously reported levels in HeLa cells (*Kjellberg et al., 2014*). In samples treated with fusogenic liposomes, the incorporated Bodipy-lipids could be specifically detected (*Figure 1—figure supplement 3A and B*). The incorporated levels of Bodipy-LacCer and Bodipy-SM $C_{12}$ per 400 000 cells were measured to be 4.2 pmol and 2.7 pmol, respectively, (i.e., $6.3 \times 10^6$ and $4.0 \times 10^6$ lipids/cell) (*Figure 1D*). To assess the incorporation efficiency of Chol, deuterium-labeled Chol, d7-Chol, was included in fusogenic liposomes. GC-MS/MS analysis revealed that d7-Chol was incorporated to similar levels as Bodipy-labeled LacCer and SM $C_{12}$. Given that the PM of these HeLa cells harbor around $7 \times 10^9$ lipids/cell (see Materials and methods section for details), the levels of Bodipy- and d7-labeled lipids detected by mass spectrometry led to a 0.02–0.09% increase in specific labeled lipids and a 0.4–1.6% of total lipids.

To determine the rate of incorporation of the different Bodipy-labeled lipid species, we used spinning disk microscopy in a central confocal plane of the cell (*Figure 1E*). Quantitative analysis of lipid incorporation into the PM over time revealed similar levels for most lipids ranging from 1900 to 4800 arbitrary units at 10 min (*Figure 1E*). The variation in incorporation rates between the different lipids may result from differences in the fusogenicity of the liposomes or differences in the PM-turnover of each particular lipid. To monitor the lateral diffusion of the Bodipy-lipids within the PM, cells were incubated with fusogenic liposomes, and fluorescence recovery after photobleaching (FRAP) in a defined ROI was monitored. All lipids showed similar smooth and rapid recovery after bleaching (*Figure 1F*). Although, as previously observed (*Kang et al., 2019*), the diffusion rate and mobile fraction of specific lipids displayed slight variations when measured by FRAP in living cells.

To conclude, the use of fusogenic liposomes enabled rapid incorporation of approximately $4 \times 10^6$ specific lipids into the PM per living cell over 10 min. To put this into context, we estimated that

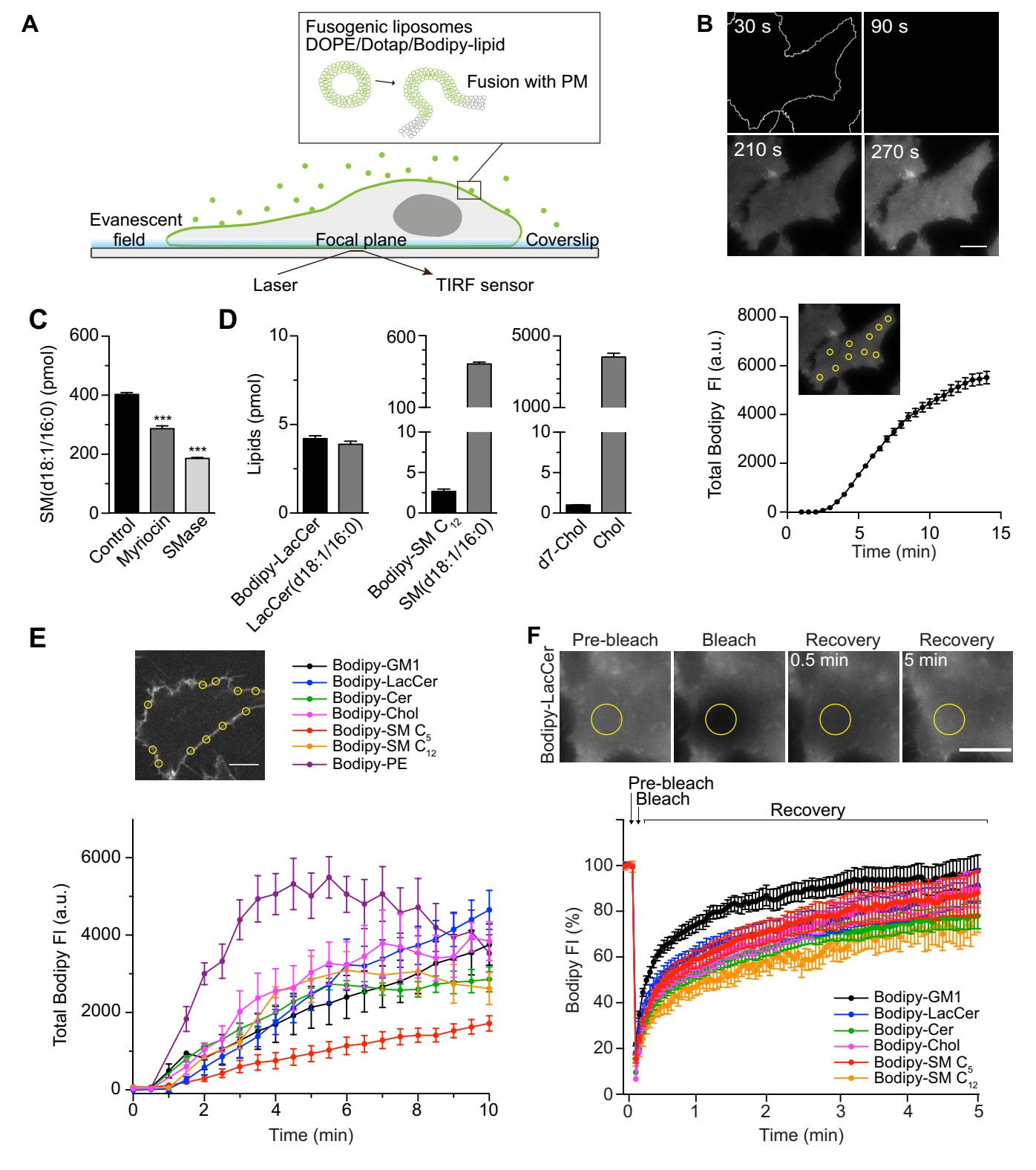

**Figure 1.** Rapid insertion of Bodipy-labeled lipids into the PM of living cells using fusogenic liposomes. (**A**) Fusogenic liposomes are used to insert Bodipy-labeled lipids into the PM. Their rapid distribution is followed in real time using TIRF microscopy. (**B**) Image sequence of Bodipy-LacCer distribution throughout basal membrane of HeLa cells. Total Bodipy fluorescence intensity (FI) was measured within ROIs (yellow insert) using Zeiss Zen interface. $n = 10$, three independent experiments, mean ± SEM. (**C**) Quantification of endogenous SM(d18:1/16:0) using LC-ESI-MS/MS in control cells

*Figure 1 continued on next page*

*Figure 1 continued*

or cells treated with SMase or myriocin for 2 hr or 24 hr, respectively. Data are shown as mean ± SD. ***, p≤0.001 vs. control. (D) Quantification of Bodipy- or d7-labeled lipids (black bars) and endogenous lipids (grey bars) in cells following incubation of cells with fusogenic liposomes. Analysis was performed using mass spectrometry. Data are shown as mean + SD. (E) Incorporation rate of Bodipy-lipids into PM of live cells. HeLa cells were treated with fusogenic liposomes (final total lipid concentration 7 nmol/ml). Total Bodipy fluorescence intensity (FI) was measured within circular ROIs (see insert) in a confocal section using spinning disk microscopy. Ten ROIs were analyzed using the Zeiss Zen system software. $n \geq 2$, two independent experiments, mean ± SEM. Scale bars, 10 μm. (F) TIRF FRAP of Bodipy-lipids after incorporation into PM of HeLa cells. A circular ROI was photobleached and recovery of Bodipy FI was monitored over 5 min. Bodipy FI was normalized to background and reference. $n \geq 10$, mean ± SEM. The online version of this article includes the following source data and figure supplement(s) for figure 1:

**Source data 1.** Excel file containing source data pertaining to *Figure 1B–F*.
**Figure supplement 1.** Caveolae dynamics at the cell surface.
**Figure supplement 2.** Liposome characterization and vesicle fusion with the PM.
**Figure supplement 2—source data 1.** Excel file containing source data pertaining to *Figure 1—figure supplement 2A–C and E*.
**Figure supplement 3.** Quantification of Bodipy-lipid incorporation using mass spectrometry.
**Figure supplement 3—source data 1.** Excel file containing source data pertaining to *Figure 1—figure supplement 3A and B*.
**Figure supplement 4.** Characterization of Cav1-mCh HeLa cells.
**Figure supplement 4—source data 1.** Excel file containing source data pertaining to *Figure 1—figure supplement 4C*.

each cell contains around 300 caveolae, based on our staining and tracking data (*Figure 1—figure supplement 4C*), comprising around 0.1% of the surface area. Caveolae are approximately 60 nm in diameter, and each lipid occupies 0.62 Å. Therefore *ca.* $10 \times 10^6$ lipids are contained within the caveolae, of which 50% is Chol. This means that the amount of specific incorporated lipids in our system is about half of the total amount of lipids contained within caveolae. The immediate addition of extra lipids to the PM did not result in a detectable effect on the cell volume (*Figure 1—figure supplement 2E*).

## Single particle tracking reveals caveolae dynamics in living cells

We next aimed to elucidate whether lipids are involved in controlling the balance between stable and dynamic caveolae at the PM, and if effects could be attributed to individual lipid species. To visualize caveolae, we generated a stable mammalian Flp-In T-Rex HeLa cell line expressing Cav1-mCherry, hereafter named Cav1-mCh HeLa cells. Expression of Cav1-mCherry was induced by doxycycline (Dox) at endogenous Cav1 levels, resulting in similar caveolae numbers to those without induction (*Figure 1—figure supplement 4A–C*). Using TIRF microscopy and single-particle tracking, we determined the time each Cav1-mCh positive punctuate structure spent at the PM (track duration) and the speed of an object (track mean speed) in, or close to, the PM (see Materials and method section for detailed tracking parameters and *Figure 2—figure supplement 3*). Given the previously reported surface dynamics of caveolae (*Pelkmans and Zerial, 2005*; *Boucrot et al., 2011*; *Mohan et al., 2015*), we postulated that stable caveolae will have a long duration and low speed, limited by their lateral diffusion in the PM (*Figure 2A*, 'Stable'). Caveolae that scission off or re-fuse with the PM during the recording period will give rise to shorter mean duration and increased mean speed. Caveolae that remain close to the surface and undergo rounds of scission and fusion, will result in an overall increase in tracks (*Figure 2A*). Caveolae disassembly will give rise to shorter tracks but no increase in mean speed and importantly a major loss of tracks. We did indeed observe a clear correlation between the track duration and track mean speed where, in general, short tracks exhibited higher speeds, whereas long tracks displayed lower speeds (*Figure 2B* and *Figure 2—figure supplement 1A*). Although the numbers of caveolae in each cell were similar at the beginning and end of the recording, we found that the number of tracks far exceeded the caveolae numbers (*Figure 2—figure supplement 2B*). This was expected as surface adjacent caveolae would give rise to several tracks. However, a drop in the fluorescent signal just below the set threshold value, would also contribute to a divided track, resulting in an overestimation of short tracks versus long tracks. Therefore, we did not consider the average duration and speed as absolute, but rather used them to compare between experimental runs with differing conditions. To verify that the tracking was sensitive to differences in caveolae dynamics, we depleted cells of EHD2, which has been shown to stabilize caveolae to the cell surface (*Morén et al., 2012*; *Stoeber et al., 2012*). Particle tracking analysis showed that the pool of tracks with high speed increased, while the

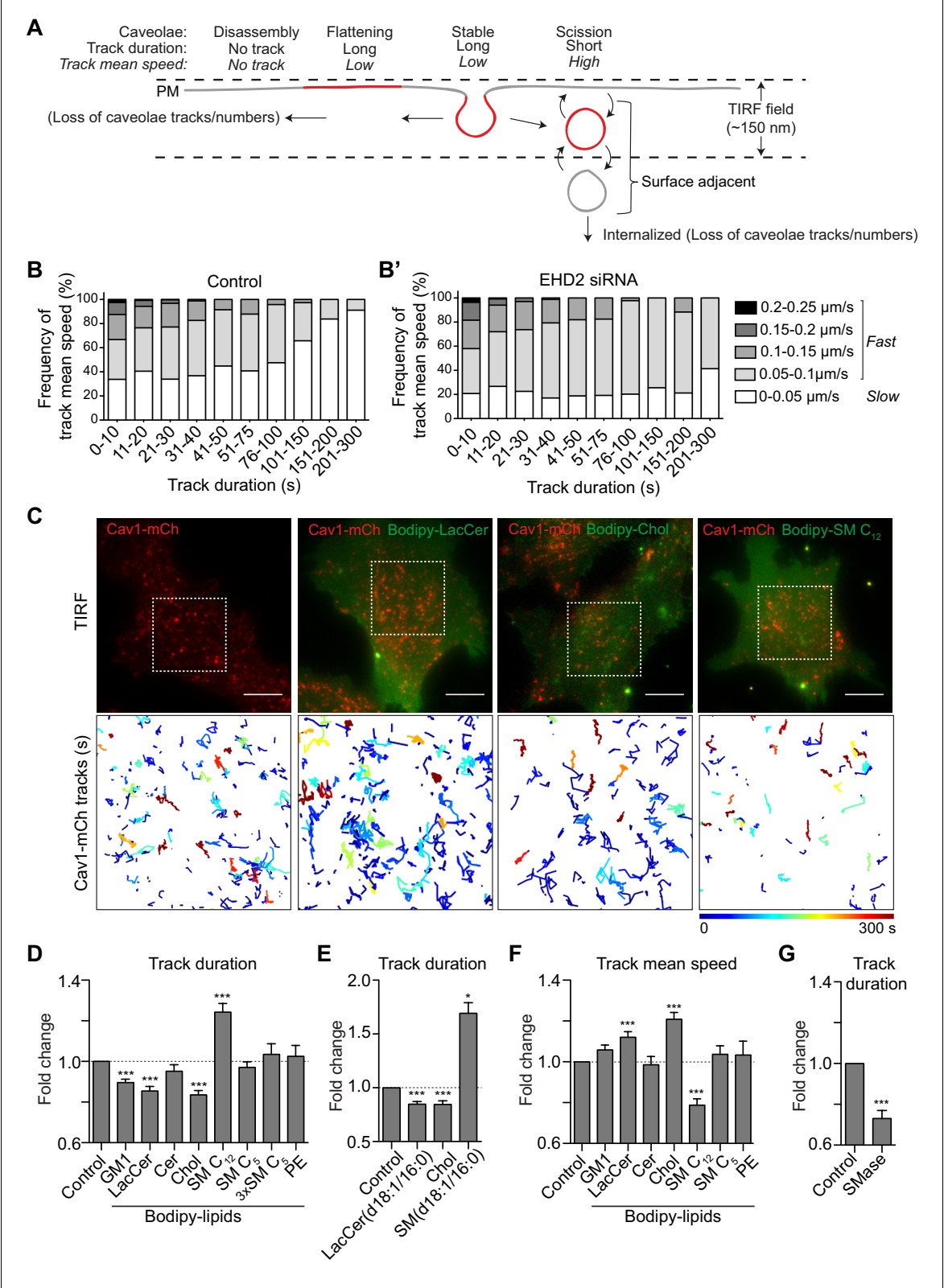

**Figure 2.** GSLs and Chol decrease the surface stability of caveolae. (**A**) Scheme showing different dynamic behaviors of caveolae. (**B, B'**) Distribution of track mean speed among subpopulations of track duration of Cav1-mCh structures (**B**) and after EHD2 depletion (**B'**). Five datasets for each condition were analyzed from TIRF live cell movies. (**C**) Representative images from TIRF movies of Cav1-mCh HeLa cells and after 15 min incubation with liposomes containing Bodipy-lipids. Color-coded trajectories illustrate time that structures can be tracked at PM over 5 min (dotted square). Scale bars,

*Figure 2 continued on next page*

*Figure 2 continued*

10 µm. See *Figure 2—videos 1*, *2*, *3*, *4*. (D,E) Quantification of track duration of Cav1-mCh structures from TIRF movies after incubation with liposomes containing labeled (D) or unlabeled lipids (E). Fold changes are relative to control (Cav1-mCh). (D) $n \geq 8$, at least two independent experiments; (E) $n \geq 8$, two independent experiments, ***, p≤0.001 vs. control. (F) Quantification of track mean speed of Cav1-mCh structures from TIRF movies (same cells as in (D)). (G) Quantification of track duration of Cav1-mCh structures from TIRF movies following incubation with SMase for 2 h. Fold changes are relative to control (Cav1-mCh). $n \geq 5$. All analyses were performed using Imaris software and data are shown as mean ± SEM.

The online version of this article includes the following video, source data, and figure supplement(s) for figure 2:

**Source data 1.** Excel file containing source data pertaining to *Figure 2B and D–G*.
**Figure supplement 1.** Correlation between track duration and track mean speed after different treatments.
**Figure supplement 1—source data 1.** Excel file containing source data pertaining to *Figure 2—figure supplement 1A–D*.
**Figure supplement 2.** Effect of Bodipy-lipid addition on Cav1-mCh tracks and quantification of Cav1-mCh positive structures.
**Figure supplement 2—source data 1.** Excel file containing source data pertaining to *Figure 2—figure supplement 2B–D*.
**Figure supplement 3.** Caveolae tracking with Imaris software.
**Figure 2—video 1.** Cell surface dynamics of Cav1-mCh.
https://elifesciences.org/articles/55038#fig2video1
**Figure 2—video 2.** Cell surface dynamics of Cav1-mCh after treatment with Bodipy-LacCer.
https://elifesciences.org/articles/55038#fig2video2
**Figure 2—video 3.** Cell surface dynamics of Cav1-mCh after treatment with Bodipy-Chol.
https://elifesciences.org/articles/55038#fig2video3
**Figure 2—video 4.** Cell surface dynamics of Cav1-mCh after treatment with Bodipy-SM $C_{12}$.
https://elifesciences.org/articles/55038#fig2video4

pool with low speed decreased (*Figure 2B'* and *Figure 2—figure supplement 1B*). When the average track duration was considered, this translated into a 0.65 fold decrease compared to control cells (*Figure 3E*), proving that the particle tracking was indeed sensitive enough to register caveolae scission induced by removal of EHD2.

## Caveolae surface stability is influenced by distinct lipid composition

Next, we screened the influence of different lipid species on caveolae mobility at the PM using tracking. To do this, fusogenic liposomes loaded with relevant lipids were added to Cav1-mCh HeLa cells, and TIRF movies were recorded immediately (*Figure 2C*). PE was used in control liposomes as it is abundant in the PM. Following incorporation of PE, caveolae dynamics remained unchanged, showing that the fusion of liposomes did not obstruct caveolae dynamics (*Figure 2D* and *Figure 2—figure supplement 2A*). In comparison to controls, Bodipy-labeled GSLs (GM1 and LacCer) and Chol significantly reduced the lifetime of caveolae at the cell surface as indicated by decreased track duration and increased mean speed (*Figure 2C,D,F*, *Figure 2—figure supplement 2A*, *Figure 2—videos 1*, *2*, *3*). For example, LacCer treatment increased the number of fast moving and short-lived caveolae versus slow moving and long-lived species, which was comparable to the EHD2 depletion (*Figure 2B'*, *Figure 2—figure supplement 1B–D* and *Figure 2—figure supplement 2C*).

A direct comparison between LacCer and Cer revealed that Cer did not enhance caveolae dynamics in a similar fashion (*Figure 2D and F* and *Figure 2—figure supplement 2A*). No difference was observed in the number of caveolae present in the PM before and after the addition of liposomes (*Figure 2—figure supplement 2C*), showing that caveolae were not disassembled and indicating that most of the scissioned caveolae remained surface adjacent. To verify that the effect was not an artifact of the Bodipy label, we treated cells with liposomes containing either unlabeled LacCer [(LacCer(d18:1/16:0)], unlabeled Chol or unlabeled SM [(SM(d18:1/16:0)] and quantified the track duration. This showed that the unlabeled lipids had the same effect on the caveolae dynamics as the corresponding Bodipy-labeled analogues (*Figure 2E*). Interestingly, when cells were treated with unlabeled SM or Bodipy-SM $C_{12}$, most of the caveolae were stable at the PM. This was characterized by a dramatic increase in track duration, and a reduction of the track mean speed (*Figure 2D–F*). However, incorporation of SM $C_5$ did not significantly alter caveolae dynamics, suggesting that the fatty acid chain length was important for this effect. Even at three times the amount (3x SM $C_5$) to control for incorporation rate, no effect was observed (*Figure 2D* and *Figure 2—figure supplement 2D*). To further investigate the role of SM, we analyzed caveolae duration following

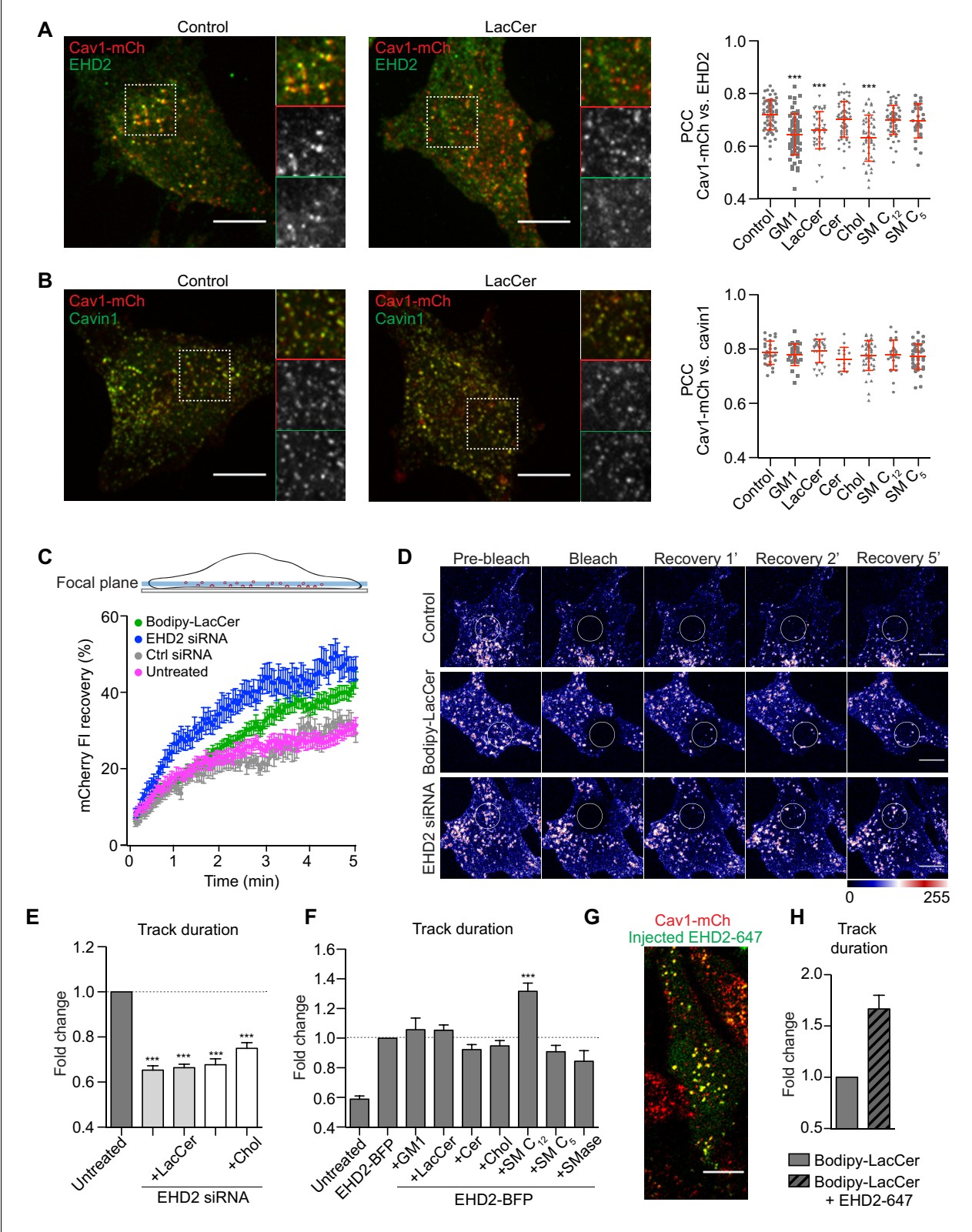

**Figure 3.** Chol and GSLs induce surface release of caveolae via an EHD2-dependent mechanism. (**A**) Representative images of maximum projected confocal z-stacks of Cav1-mCh HeLa cells. Untreated cells or cells treated with LacCer-Bodipy liposomes for 1 h, fixed and immunostained for endogenous EHD2. High-magnification images (dotted square) show localization of EHD2 to Cav1-mCh (see scatterplot for quantification). $n \geq 60$, two independent experiments, mean ± SEM. ***, p≤0.001 vs. control. (**B**) Experimental protocols analogous to (**A**), with exception of endogenous cavin1

*Figure 3 continued on next page*

*Figure 3 continued*

immunostaining. $n \geq 60$, mean ± SEM. (C) Confocal FRAP of Cav1-mCh HeLa cells treated with either EHD2 siRNA or Bodipy-LacCer liposomes. A ROI was photobleached and recovery of mCherry FI monitored over 5 min. mCherry FI was normalized to background and reference. $n \geq 10$, mean ± SEM. (D) Representative time-lapse series showing control Cav1-mCh HeLa cells and cells treated with either EHD2 siRNA or Bodipy-LacCer liposomes. The photobleached area is outlined with white circles. mCherry FI is intensity-coded using LUT. (E) Effects of lipids on track duration of Cav1-mCh structures were analyzed following siRNA-mediated depletion of EHD2. $n \geq 8$, two independent experiments, mean ± SEM. (F) Quantification of track duration of untreated Cav1-mCh HeLa cells or cells transiently expressing EHD2-BFP with or without incubation with liposomes. Changes in track duration are relative to EHD2-BFP control (indicated by dotted line). $n \geq 8$, two independent experiments, mean + SEM. ***, $p \leq 0.001$ vs. control cells. (G) Representative live cell confocal image of EHD2-647 microinjected into Cav1-mCh HeLa cells. (H) Quantification of track duration of Cav1-mCh cells treated with Bodipy-LacCer and following microinjection of EHD2-647. $n = 8$, mean + SEM. All scale bars, 10 µm.

The online version of this article includes the following source data and figure supplement(s) for figure 3:

**Source data 1.** Excel file containing source data pertaining to *Figure 3A–C,E,F and H*.
**Figure supplement 1.** Cavin1 localization to Cav1-mCh before and after liposome treatment.
**Figure supplement 1—source data 1.** Excel file containing source data pertaining to *Figure 3—figure supplement 1B*.
**Figure supplement 2.** EHD2 depletion in Cav1-mCh HeLa cells.
**Figure supplement 2—source data 1.** Excel file containing source data pertaining to *Figure 3—figure supplement 2B and C*.
**Figure supplement 3.** Stabilization of caveolae to the PM by EHD2-I157Q cannot be reversed by addition of Bodipy-labeled LacCer or Chol.
**Figure supplement 3—source data 1.** Excel file containing source data pertaining to *Figure 3—figure supplement 3B,D and E*.
**Figure supplement 4.** Microinjection of EHD2-647 into Cav1-mCh HeLa cells.
**Figure supplement 4—source data 1.** Excel file containing source data pertaining to *Figure 3—figure supplement 4B*.

SMase treatment, and found that this resulted in a decreased track duration, in agreement with a surface-stabilizing role for this lipid (*Figure 2G*).

## Chol and GSLs induce scission of caveolae via mechanisms that are counteracted by EHD2

EHD2 normally localizes with the majority of surface associated caveolae (*Morén et al., 2012*). We aimed to address whether the increased caveolae dynamics induced by either Chol or GSLs were caused by their PM release, as characterized by loss of the stabilizing protein EHD2. Therefore, we treated Cav1-mCh HeLa cells with the fusogenic liposomes and visualized endogenous EHD2 using indirect immunofluorescent labeling (*Figure 3A*). These experiments revealed that incorporation of GM1, LacCer, or Chol into the PM led to a significantly lower amount of EHD2 that colocalized with Cav1 (*Figure 3A*, scatter plot). These data suggest that the caveolae release induced by increased PM levels of LacCer and Chol is results from loss of EHD2-mediated stabilization. Conversely, Cer and SM $C_{12}$, as well as its short chain analogue SM $C_5$, did not appear to have any significant effect on the association of EHD2 with Cav1 (*Figure 3A*, scatter plot). Furthermore, following lipid treatment, the majority of caveolae remained associated with cavin1 (*Figure 3B*). To verify that caveolae at the cell surface were not disassembled via detachment of cavin1, we compared the colocalization of Cav1-mCh with cavin1-GFP before and after lipid treatment using TIRF microscopy (*Figure 3—figure supplement 1A–B*). These data showed that there was no significant change in the number of cavin-decorated caveolae following incorporation of any of the specific lipids, revealing that no disruption of the caveolae coat, and subsequent release of cavin1 occurred.

As increased scission of caveolae from the cell surface results in more mobile intracellular caveolar vesicles (*Stoeber et al., 2012*), we performed fluorescence recovery after photobleaching (FRAP) experiments to investigate the recovery rate of caveolae. Addition of LacCer resulted in more mobile caveolae inside the cells in comparison with control cells (*Figure 3C–D*). The recovery rate after LacCer addition was similar to the rate in EHD2-depleted cells (see *Figure 3—figure supplement 2A* for protein levels). This further supported the hypothesis that lipid incorporation drives caveolae scission. When LacCer or Chol were added to EHD2-depleted cells we found that the caveolae track duration was not further reduced in comparison with EHD2-depleted cells (*Figure 3E* and *Figure 3—figure supplement 2B* for control siRNA). These results supported our hypothesis that the lipid-induced effect on caveolae dynamics resulted from loss of surface stability mediated by EHD2, and implied that EHD2 controls the stability of caveolae in response to lipid composition.

To test whether increased levels of EHD2 could restore caveolae stability after lipid treatment, we transiently expressed blue fluorescent protein (BFP)-tagged EHD2 in Cav1-mCh HeLa cells. Analysis

of TIRF live cell movies showed that EHD2-BFP positive caveolae were highly stable compared to untreated cells (*Figure 3F*). In the presence of EHD2-BFP, the destabilizing effect seen for GM1, LacCer, and Chol was abolished, as demonstrated by negligible changes in track duration compared to control conditions (*Figure 3F*). This suggested that excess levels of EHD2 were capable of restricting the effect of excess Chol and GSLs. In addition, tracking of caveolae in cells stably expressing Cav1-mCh and EHD2-BFP showed that most caveolae were positive for EHD-BFP (96%) and that this population was more stable than the population lacking EHD2-BFP (4%) (*Figure 3—figure supplement 3A–C*). Surprisingly, SM $C_{12}$ had an additive effect and led to predominantly stable caveolae (*Figure 3F*), suggesting that here the stabilizing effect of SM is increased in the presence of excess levels of EHD2. Furthermore, EHD2 could partly restrict the increase in caveolae dynamics observed in response to SMase treatment and decreased SM levels in the PM. These data suggested that SM and EHD2 both stabilize caveolae, but perhaps via alternate mechanistic pathways.

To test whether EHD2-mediated suppression of the lipid effect relied on multiple rounds of EHD2 assembly and disassembly at caveolae, we overexpressed a BFP-tagged ATP-cycle mutant, EHD2-I157Q. The increased ATP hydrolysis rate of this mutant leads to stable association of EHD2-I157Q to caveolae and a slower exchange rate (*Figure 3—figure supplement 3D*; *Daumke et al., 2007*; *Stoeber et al., 2012*; *Hoernke et al., 2017*). We observed that, when co-expressed in Cav1-mCh cells, both EHD2-I157Q and EHD2 stabilized caveolae at the PM to similar extents, independent of treatment with either LacCer or Chol (*Figure 3—figure supplement 3E*). This verified that stable assembly, but not disassembly of EHD2, is necessary to stabilize caveolae.

To clarify whether, in order to have a stabilizing role, EHD2 had to be caveolae-associated prior to lipid addition, fluorescently labeled, purified EHD2 (EHD2-647) was microinjected into Cav1-mCh HeLa cells (*Figure 3G*). Within 20 min, EHD2-647 colocalized with Cav1, confirming that the microinjected protein was indeed recruited to caveolae (*Figure 3—figure supplement 4A–B*). Next, we tested whether an acute injection of EHD2-647 could rescue the effect on caveolae dynamics caused by LacCer. Strikingly, we found that exogenously added EHD2 stabilized the caveolae to the same extent as the overexpressed EHD2, demonstrating that increased levels of EHD2 can acutely reverse the increased mobility of caveolae induced by lipids (*Figure 3H*).

## Chol and SM are sequestered within caveolae

As SM $C_{12}$ increased the surface stability of caveolae, while GSLs and Chol promoted scission, we investigated whether there was a differential accumulation of these lipids within caveolae at the PM. We treated Cav1-mCh HeLa cells with fusogenic liposomes and followed the distribution of Bodipy-labeled lipids using live cell TIRF microscopy. After 15 min, LacCer, Chol , and SM $C_{12}$ lipids were found to colocalize with Cav1-mCh positive structures (*Figure 4—figure supplement 2A–D* and *Figure 4—videos 1* and *2*). Data analysis was hindered by high caveolae mobility following addition of LacCer and Chol, and the extent of colocalization could not be quantified. To circumvent this, we overexpressed EHD2-BFP to stabilize caveolae at the PM. Interestingly, nearly 80% of caveolae positive for EHD2 were also positive for LacCer, Chol, and SM $C_{12}$ (*Figure 4A–C*, *Figure 4—figure supplement 1A–B* and *Figure 4—videos 3* and *4*). In comparison, Cer, which had no effect on caveolae dynamics, did not localize to caveolae, even in the presence of EHD2-BFP (*Figure 4D*, *Figure 4—figure supplement 1A* and *Figure 4—figure supplement 2D*).

To investigate lipid exchange between the stable caveolae and the surrounding PM, we performed FRAP experiments. The Bodipy-LacCer signal reappeared rapidly at precisely the bleached spot positive for Cav1-mCh, with a close to quantitative fluorescence recovery (*Figure 4E–F*). This indicated that the lipid diffused freely throughout the PM and, following photobleaching, re-accumulated quickly within caveolae. In comparison, Bodipy-Chol recovered much slower with 60% of the initial signal being restored after 5 min acquisition time (*Figure 4F* and *Figure 4—figure supplement 3A*). This showed that there is a large immobile pool of Chol in caveolae that is sequestered from the rest of the PM. The signal from Bodipy-SM $C_{12}$ was almost exclusively detected in caveolae hampering recovery measurements outside of this domain (*Figure 4—figure supplement 3B–C*). However, the recovery rate of the SM $C_{12}$ within caveolae was very low, showing that SM $C_{12}$ is also sequestered in caveolae in the presence of EHD2. Our data suggest that LacCer, Chol, and SM $C_{12}$ are highly enriched in caveolae and, while the lateral diffusion of LacCer in and out of caveolae is high, Chol and SM $C_{12}$ are restrained to this invagination.

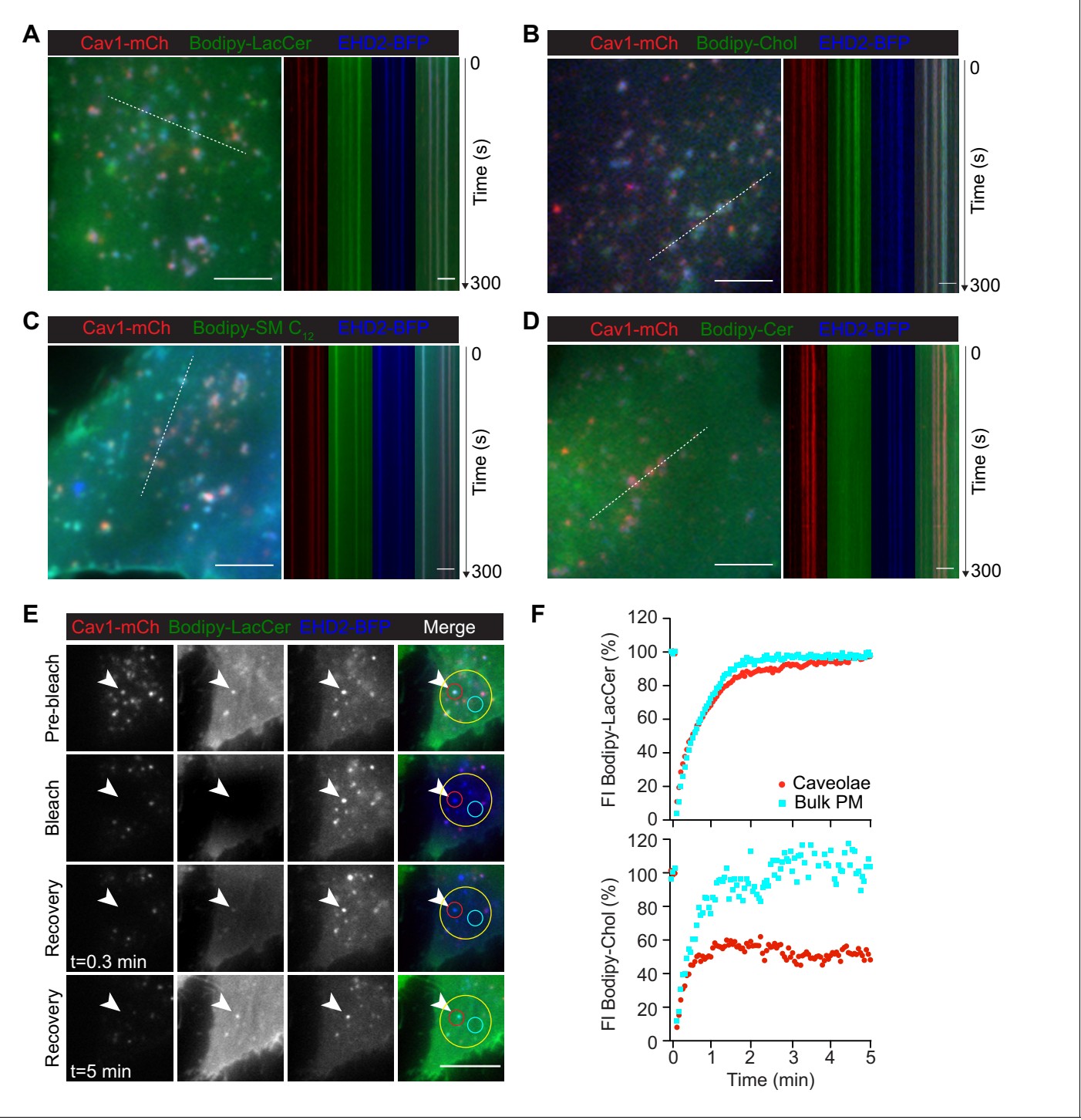

**Figure 4.** LacCer, Chol, and SM accumulate in caveolae and Chol is sequestered within these domains. (**A–D**) Cav1-mCh HeLa cells transiently expressing EHD2-BFP were incubated with Bodipy-lipids. White lines indicate location of kymograph. TIRF movies were recorded at 3 s intervals for 5 min. Scale bars, 10 μm; kymograph scale bars, 5 μm. See *Figure 4—videos 3* and *4*. (**E**) Cav1-mCh HeLa cells transiently expressing EHD2-BFP were incubated with Bodipy-LacCer for 10 min. Following photobleaching (yellow ROI), recovery of Bodipy signal within caveolae (red ROI) and bulk PM (cyan ROI) was monitored over time. White arrows highlight surface connected caveolae with accumulated Bodipy-LacCer. Scale bar, 10 μm. (**F**) Recovery curves of Bodipy intensities within caveolae ROI (red) and bulk PM ROI (cyan). Bodipy FI was normalized to background and reference. The online version of this article includes the following video, source data, and figure supplement(s) for figure 4:

**Source data 1.** Excel file containing source data pertaining to *Figure 4F*.

**Figure supplement 1.** Localization of Bodipy-lipids to Cav1-mCh positive for EHD2-BFP.

*Figure 4 continued on next page*

*Figure 4 continued*

**Figure supplement 1—source data 1.** Excel file containing source data pertaining to *Figure 4—figure supplement 1A and B*.
**Figure supplement 2.** Kymographs of Cav1-mCh HeLa cells incubated with Bodipy-lipid liposomes.
**Figure supplement 2—source data 1.** Excel file containing source data pertaining to *Figure 4—figure supplement 2A–D*.
**Figure supplement 3.** Bodipy-Chol and Bodipy-SM $C_{12}$ are sequestered within caveolae.
**Figure supplement 3—source data 1.** Excel file containing source data pertaining to *Figure 4—figure supplement 3C*.
**Figure 4—video 1.** Bodipy-LacCer colocalizes with Cav1-mCh positive structures.
https://elifesciences.org/articles/55038#fig4video1
**Figure 4—video 2.** Bodipy-Chol colocalizes with Cav1-mCh positive structures.
https://elifesciences.org/articles/55038#fig4video2
**Figure 4—video 3.** Bodipy-LacCer accumulates in caveolae.
https://elifesciences.org/articles/55038#fig4video3
**Figure 4—video 4.** Bodipy-Chol accumulates in caveolae.
https://elifesciences.org/articles/55038#fig4video4

## Chol accumulation reduces the caveolae diameter in 3T3-L1 adipocytes

To elucidate whether lipid accumulation affected the overall morphology of surface connected caveolae in Cav1-mCh HeLa cells, we analyzed their ultrastructure in cells overexpressing EHD2. As the number of caveolae in the PM of these cells is relatively low, we used correlative light electron microscopy (CLEM) to specifically identify fluorescently tagged caveolae by combining light microscopy with the higher resolution images of transmission electron microscopy (TEM). Fluorescence light microscopy images of Cav1-mCh HeLa cells were superimposed with correlative electron micrographs to find the closest match of fluorescence signal to structure using the nuclear stain as a guide (*Figure 5—figure supplement 1A*). The surface connected caveolae in cells treated with LacCer, Chol, and SM $C_{12}$, displayed a similar flask-shaped morphology to that seen in control cells, verifying that lipid addition did not majorly distort caveolae morphology (*Figure 5A* and *Figure 5—figure supplement 1B*).

To quantitatively assess differences in morphology related to caveolae scission in a more physiologically relevant system, we used adipocytes as they are a main source of cholesterol storage and efflux (*Krause and Hartman, 1984*), but are devoid of LacCer (*Ortegren et al., 2004*). We differentiated 3T3-L1 cells to adipocytes, which results in upregulation of Cav1 and EHD2 (*Figure 5—figure supplement 2A*; *Morén et al., 2019*), and formation of a large number of caveolae (*Thorn et al., 2003*) that could be clearly distinguished from clathrin-coated pits (*Figure 5—figure supplement 2B*). Lipid incorporation quantification verified that fusogenic liposomes could be used to insert specific lipids into the PM of these cells (*Figure 5—figure supplement 2C*). Using TEM, we analyzed the dimensions of caveolae before and after Chol addition (*Figure 5B–E*). We found that the neck diameter of surface associated caveolae were significantly decreased and more homogeneous following Chol incorporation in comparison to control cells (*Figure 5D*). Furthermore, the bulb width was also significantly smaller resulting in more drop-shaped caveolae (*Figure 5D′*). Quantitative analysis of the caveolae population where a clear surface connected neck could not be detected allowed measurement of the surface area of caveolae. Comparison to control cells showed that area, as well as bulb width, decreased following Chol addition (*Figure 5E–E′*). Furthermore, Chol incorporation resulted in a more homogeneous caveolae population in terms of size and dimensions. These data suggested that an acute increase in Chol levels in the PM of 3T3-adipocytes induced alterations in the caveolae coat architecture, resulting in reduced neck diameter and a smaller more uniform bulb diameter.

## GSLs are internalized to the endosomal system independently of Cav1, while Chol is predominantly trafficked to lipid droplets

Next, we aimed to address whether caveolae scission significantly contributed to internalization and trafficking of lipids in our system as previously proposed (*Puri et al., 2001*; *Le Lay et al., 2006*; *Shvets et al., 2015*). We used fusogenic liposomes to investigate whether Bodipy-labeled LacCer or Chol were internalized and trafficked through the endosomal pathway following incorporation into the PM. To mark early endosomes (EE), Rab5-BFP was transiently expressed in Cav1-mCh HeLa cells.

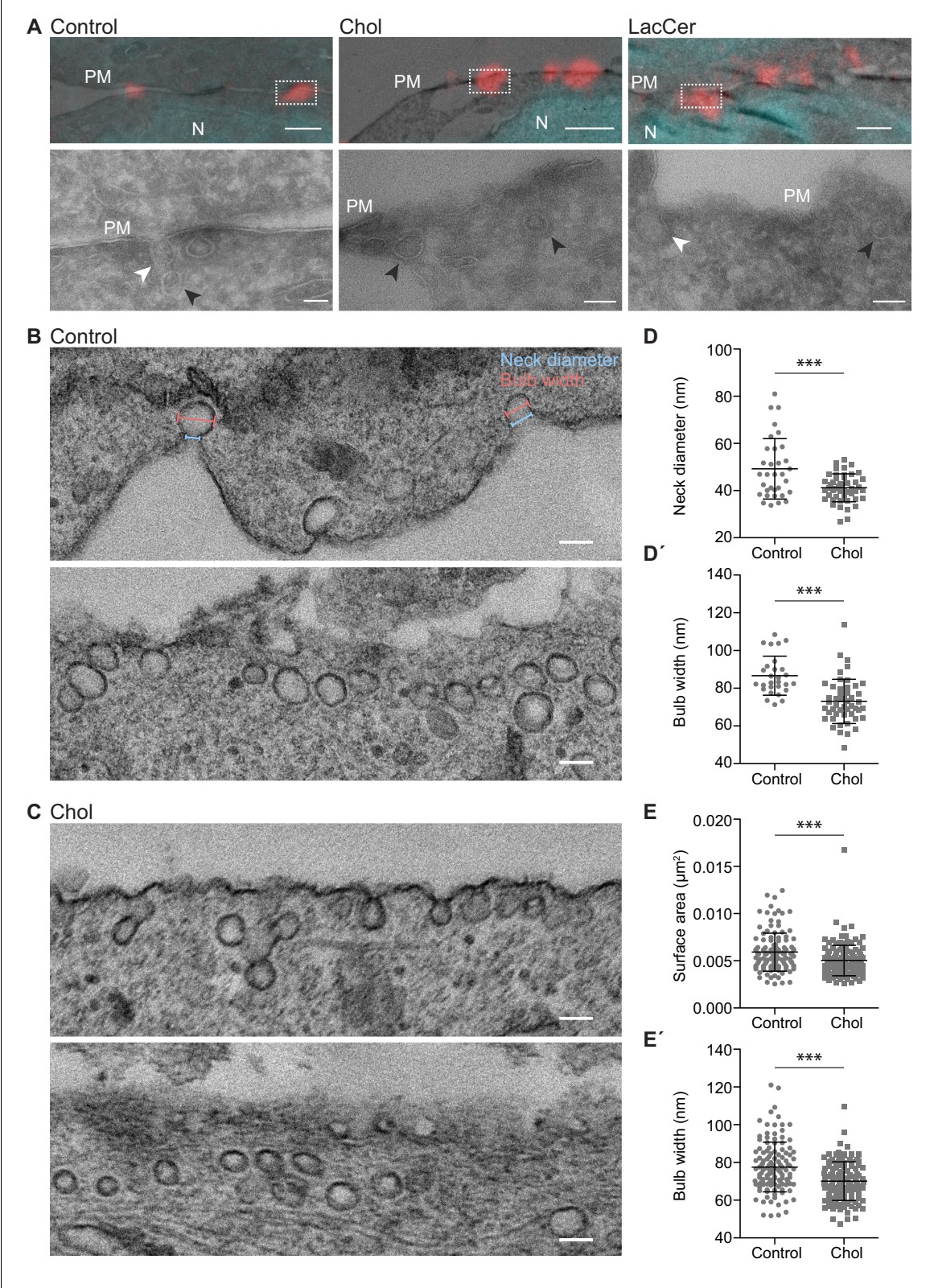

**Figure 5.** Chol accumulation reduces the caveolae diameter in 3T3 adipocytes. (**A**) Representative overlays of light microscopy images with corresponding electron micrographs showing localization of caveolae (Cav1-mCh in red) and nuclei (DAPI in cyan) for untreated Cav1-mCh HeLa cells (control) or cells treated with Bodipy-labeled Chol or LacCer. Dotted boxes show regions of higher magnification in corresponding panels below. N, nucleus; PM, plasma membrane. White arrows denote surface connected caveolae and black arrows indicate surface adjacent caveolae. Scale bars, 1

*Figure 5 continued on next page*

*Figure 5 continued*

μm; inset scale bars, 100 nm. (**B, C**) Electron micrographs of control 3T3-L1 adipocytes (**B**) and 3T3-L1 adipocytes treated with Bodipy-Chol (**C**). Top and bottom panels show two representative images per sample: one with clear surface connected necks and one without. Cells were chemically fixed, embedded in resin, and processed for electron microscopy. Scale bars, 100 nm. (**D, D′**) Scatter plots showing the quantification of neck diameter (**D**) and bulb width (**D′**) of surface connected caveolae in 3T3-L1 adipocytes. Bulb width and neck diameter are highlighted in (**B**), upper panel. $n \geq 30$, mean ± SEM. (**E, E′**) Scatter plots showing the quantification of surface area (**E**) and bulb width (**E′**) of surface adjacent caveolae in 3T3-L1 adipocytes. $n \geq 120$, mean ± SEM. \*\*\*, $p \leq 0.001$.

The online version of this article includes the following source data and figure supplement(s) for figure 5:

**Source data 1.** Excel file containing source data pertaining to *Figure 5D and E*.
**Figure supplement 1.** Correlative electron microscopy of caveolae.
**Figure supplement 2.** Expression of EHD2 and Cav1 is upregulated in 3T3-L1 adipocytes.
**Figure supplement 2—source data 1.** Excel file containing source data pertaining to *Figure 5—figure supplement 2C*.

Cells were incubated with fusogenic liposomes for either 15 min or 3 h, followed by fixation and EE localization was quantified. We observed localization of LacCer to the EE but not to the Golgi, contrasting previous studies using BSA-Bodipy-LacCer (*Puri et al., 2001*). After 15 min, more than half of the EE were positive for LacCer (55%) compared to only 6% for Chol (*Figure 6A–B*). After 3 h, the number of LacCer-positive EE remained constant, whereas the EE positive for Chol had increased to 18% (*Figure 6B*). The amount of caveolae that colocalized with EE was very low, both before and after treatment with lipids (*Figure 6C*). To test if caveolae were involved in lipid trafficking to the EE, the experiments were repeated in cells depleted of Cav1 (*Figure 6D–F*). After 15 min incubation time, 55% and 10% of EE were positive for LacCer and Chol, respectively (*Figure 6D–E*). This suggested that while caveolae did not seem to influence the efficiency of LacCer or Chol trafficking to endosomes, loss of Cav1 resulted in an increased amount of Chol accumulating in this compartment. Our data indicate that caveolae serve as buffers or sensors of GSL and Chol concentrations rather than endocytic vesicles.

During our experiments, we noticed that a large fraction of Chol localized to compartments distinct from the endosomal system. To determine whether Chol localized to lipid droplets (LD) as previously proposed (*Le Lay et al., 2006*; *Shvets et al., 2015*), we incubated HeLa cells with fusogenic liposomes and visualized LD using LipidTOX-DR. On average, 85% of LDs were positive for Chol after both 15 min and 3 h (*Figure 6G–H*), and similar levels of Chol-positive LD were detected in cells lacking Cav1 (*Figure 6I*). While Chol extensively localized to LD, we did not observe LacCer associated with LD (*Figure 6G–I*). These data are consistent with the hypothesis that excess Chol in the PM is trafficked directly to LD in a process that does not require caveolae per se, and that the levels of Chol taking an alternative route to EE increase in the absence of caveolae.

## Discussion

While PM turnover is typically regulated in a tightly controlled manner, marginal changes in its composition are associated with severe diseases such as cancer, diabetes, and Alzheimer's disease (*Harayama and Riezman, 2018*). It has been proposed that caveolae play a major role in preserving lipid homeostasis via sensing and buffering PM properties (*Pilch and Liu, 2011*; *Parton and del Pozo, 2013*). However, studies detailing how lipid composition influences cellular phenotypes have been hindered by a lack of methods to selectively manipulate the PM lipid composition; especially with regard to introducing specific lipids. To address this, we applied an approach for studying these systems in living cells that employs DOPE/DOTAP-based liposomes capable of mediating highly effective fusion processes with cell membranes to deliver their lipid cargoes. Such liposomes have previously been used as nanocarriers to deliver intracellular proteins (*Kube et al., 2017*). Our methodology successfully delivered specific lipids into the PM bilayer of living cells with high efficiency. These rapid fusion events have allowed us to report the first studies observing caveolae response to acute changes in PM lipid composition, as well as lipid exchange itself in the caveolae bulb. Furthermore, the use of labelled lipids allowed us to measure their incorporation relative to endogenous levels. Our results demonstrate the power of this approach for studying caveolae dynamics and we foresee that our methodology will also be a useful tool outside of this framework.

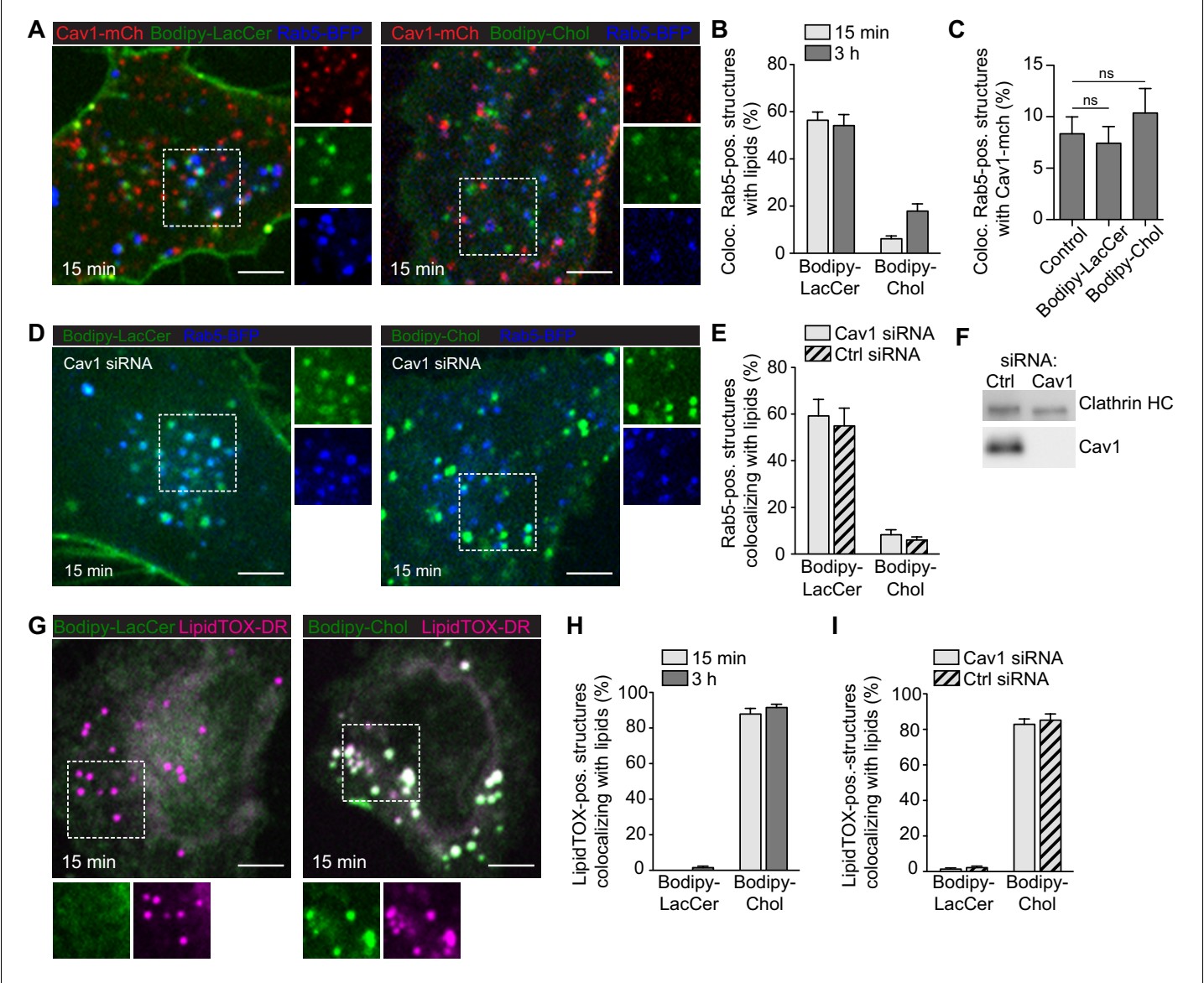

**Figure 6.** GSLs are internalized to the endosomal system independent of Cav1, while Chol is predominantly trafficked to lipid droplets. (**A**) Cav1-mCh HeLa cells expressing Rab5-BFP were incubated with Bodipy-labeled LacCer or Chol for 15 min. Individual channels are shown for selected areas (dotted box). (**B**) Colocalization of lipids with Rab5-positive structures after indicated time-points. (**C**) Quantification of Cav1-mCh localization to Rab5-BFP positive structures before (control) and after lipid addition. Statistical analysis: ns - non significant. (**D**) Cav1 siRNA-treated Cav1-mCh HeLa cells expressing Rab5-BFP after incubation with Bodipy-labeled LacCer or Chol for 15 min. High-magnification images of selected areas (dotted box) for each channel are shown. (**E**) Quantification of EE positive for lipids in cells treated with siRNA control or against Cav1. Cells were incubated with Bodipy-lipids for 15 min. (**F**) Representative immunoblots of Cav1-mCh HeLa cells treated with control siRNA or siRNA against Cav1. Clathrin HC served as loading control. (**G**) Cav1-mCh HeLa cells were incubated with Bodipy-lipids for 15 min, fixed and LDs were stained using LipidTOX-DR. (**H**) Colocalization of lipids to LDs. (**I**) Colocalization of lipids with LDs in cells depleted of Cav1 after 15 min. (**B, C, E, H, I**) n = 10, mean ± SEM. All scale bars, 5 μm.

The online version of this article includes the following source data for figure 6:

**Source data 1.** Excel file containing source data pertaining to *Figure 6B,C,E,H and I*.

Our work shows that the surface association of caveolae is highly sensitive to changes in the PM lipid composition. An acute increase in the levels of Chol and GSLs, which were found to specifically accumulate in caveolae, dramatically increased caveolae mobility. These caveolae traveled at higher speeds, their PM duration was shorter and they also displayed reduced levels of EHD2, a protein

indicative of PM-associated caveolae. Therefore, we conclude that accumulation of Chol and GSLs in caveolae triggers surface release of caveolae. In agreement with this, EM analysis revealed that the caveolae neck diameter was reduced in cells with elevated Chol levels. Our findings are consistent with previous reports suggesting that BSA-LacCer and Chol decrease the number of caveolae associated to the PM and enhance their mobility (*Sharma et al., 2004*; *Le Lay et al., 2006*). Based on the present study, increased caveolae mobility is a direct result of accumulation of these lipids in caveolae. As our methodology allowed us to determine the levels of specifically incorporated lipids, we found that rapid, yet relatively small increases in specific lipids can affect caveolae dynamics. Because caveolae immediately responded to these changes in bilayer composition, we propose that they serve as PM sensors, not only for membrane tension, but also for lipid composition.

Previous studies have suggested that a threshold concentration of Chol is required to maintain caveolae integrity and proposed that assembly and disassembly is in a dynamic equilibrium dependent on Chol levels (*Hailstones et al., 1998*). This is also in line with our experiments showing that excess Chol drives caveolae assembly towards scission and that Chol was indeed found to accumulate in caveolae when these structures were restrained to the surface by EHD2 overexpression. Furthermore, our methodology enabled, for the first time, measurement of lipid lateral flow in and out of the caveolae bulb using FRAP. Comparing the FRAP recovery of Bodipy-LacCer and Bodipy-Chol, which were both enriched in caveolae, showed that while photobleached Bodipy-LacCer was almost fully exchanged via lateral diffusion after 2 min, photobleached Bodipy-Chol was only exchanged by 50%. This showed that Chol was sequestered in caveolae, potentially through its interaction to Cav1 (*Parton and del Pozo, 2013*). In contrast, Bodipy-Cer, which lacks the disaccharide structural motif of LacCer, did not accumulate in caveolae and had no effect on their dynamics, which is in agreement with earlier reports (*Sharma et al., 2004*). Precisely how the lactosyl group mediates the caveolae-enrichment of LacCer and how this in turn drives caveolae scission is not clear.

Interestingly, we found that SM and Bodipy-SM $C_{12}$, but not Bodipy-SM $C_5$, dramatically increased caveolae stability, in terms of both speed and duration. This suggests that the chain length is important and might influence their interactions with for example Chol. SM has been shown to sequester a pool of Chol in the PM, which, together with an accessible and inaccessible pool, aid in the sensing of the Chol levels in the PM (*Das et al., 2014*). The elevated levels of SM may alter the levels of SM-sequestered Chol, thereby affecting caveolae stability. In agreement with this, we found that a dramatic reduction of SM using SMase increased caveolae dynamics. SM was highly enriched in caveolae, especially in cells with elevated EHD2 levels. This shows that this lipid has a direct role in caveolae, but the mechanism by which SM stabilizes caveolae requires more detailed investigation.

Although an area of extensive research, a consensus on the exact mechanism of caveolae scission has not yet been reached. Our observations suggest a model, where the accumulation of lipids in caveolae reduces the neck diameter, leading to scission. We speculate that this could be caused by increased access of scission-mediating molecules like dynamin to the neck, or that that these lipids promote assembly of Cav1 and cavins, which drive curvature towards scission. The lipid-driven assembly of Cav1 may be an intrinsically unstable system, eventually resulting in scission if no restraining forces are applied. This indicates that scission is tightly coupled to, and a continuum of, caveolae biogenesis. In line with this, expression of caveolin in bacterial systems induced the formation of internal caveolae-like vesicles containing caveolin so-called heterologous caveolae (*Walser et al., 2012*). The scission step could also involve lipid phase separation. A similar mechanism has previously been proposed but not experimentally validated (*Lenz et al., 2009*), and our new data show that a locally increased concentration of GSLs and Chol in caveolae may induce phase separation and therefore facilitate budding and scission of caveolae. Consistent with this, model membrane studies have shown that sterol-induced phase separation can promote membrane scission (*Bacia et al., 2005*; *Roux et al., 2005*). Of interest, in other systems GSLs and Chol have been suggested to play a crucial role in membrane nanodomain budding to generate intracellular transport carriers (*Schuck and Simons, 2004*).

EHD2 has been shown to confine caveolae to the cell surface (*Morén et al., 2012*; *Stoeber et al., 2012*). In the current study, we acutely altered the lipid composition to induce caveolae scission and analyzed the immediate role of EHD2. We found that removal of EHD2, while at the same time changing the lipid composition, did not have an additive effect on caveolae dynamics. However, excess levels of EHD2 from overexpression or direct microinjection, could suppress the

effect of the altered lipid composition. This suggests that an increased assembly rate of EHD2 at the caveolae neck is necessary and sufficient to drive the equilibrium towards stable surface association of caveolae. We conclude that oligomers of EHD2 might provide a restraining force that prevents reduction of the neck diameter and thereby inhibits phase separation or assembly of scission-mediating proteins. Similarly, EHD2 would prevent flattening of caveolae and thus, act to stabilize the typical bulb-shape of caveolae. Our data also indicate that EHD2 could limit lipid diffusion of Chol and SM over the caveolae neck thereby influencing caveolae stability. Therefore, EHD2 would act as a key regulator of caveolae dynamics in response to changes in both PM lipid composition and membrane tension.

Caveolae have been proposed to play an integral role in intracellular lipid trafficking (*Puri et al., 2001*; *Le Lay et al., 2006*; *Shvets et al., 2015*). This prompted us to examine the cellular fate of our labelled lipids. We found that while Bodipy-LacCer was internalized via the endosomal system, Chol predominately localized to LD. Importantly, and in contrast to previous data (*Shvets et al., 2015*), we found that loss of caveolae did not majorly influence the trafficking of these lipids in HeLa cells. Based on this, we propose that caveolae should not be considered as vehicles for internalization of lipids, but rather that lipid composition influences caveolae biogenesis and dynamics. Together with the caveolae coat components, it is feasible that sequestered lipids may control formation and define the size and curvature of these PM invaginations. This, together with our data showing that Chol is enriched and sequestered in caveolae, implies that caveolae could serve as reservoirs of Chol in the PM, thereby buffering the surface levels of this lipid.

Together, our findings indicate that the dynamic behavior of caveolae is highly sensitive to changes in PM lipid composition. We demonstrate that, following incorporation into the lipid bilayer, GSLs and Chol accumulate in caveolae, which promotes scission of these membrane invaginations from the cell surface. The current study redefines the fundamental understanding of how caveolae dynamics are governed by biologically relevant lipids and will be of future relevance linking caveolae malfunction with lipid disorders.

# Materials and methods

## Key resources table

| Reagent type (species) or resource | Designation | Source or reference | Identifiers | Additional information |
|---|---|---|---|---|
| Antibody | anti-PTRF (Rabbit polyclonal) | Abcam | Cat # ab48824 RRID:AB_88224 | IF(1:100) |
| Antibody | anti-EHD2 (Rabbit polyclonal) | (*Morén et al., 2012*) | RRID:AB_2833022 | IF (1:500), WB(1:2000) |
| Antibody | anti-clathrin heavy chain (clone 23; mouse monoclonal) | BD Transduction Laboratories | Cat # 610499 RRID:AB_397865 | WB (1:1000) |
| Antibody | anti-caveolin-1 (Rabbit, polyclonal) | Abcam | Cat # ab2910 RRID:AB_303405 | IF (1:500) WB (1:10 000) |
| Antibody | anti-rabbit IgG secondary antibody coupled to Alexa Fluor 647 (Goat polyclonal) | Thermo Fisher Scientific | Cat # A21246 RRID:AB_2535814 | IF (1:300) |
| Chemical compound, drug | 1,2-dioleoyl-sn-glycero-3-phosphoethanolamine (DOPE) | Avanti Polar Lipids | Cat # 850725 | |
| Chemical compound, drug | 1,2-dioleoyl-3-trimethylammonium-propane (chloride salt) (DOTAP) | Avanti Polar Lipids | Cat # 890890 | |
| Chemical compound, drug | Bodipy FL C5-Ganglioside GM1 | Thermo Fisher Scientific | Cat # B13950 | |

*Continued on next page*

Continued

| Reagent type (species) or resource | Designation | Source or reference | Identifiers | Additional information |
|---|---|---|---|---|
| Chemical compound, drug | Bodipy FL C5-LacCer | This study (*Gretskaya and Bezuglov, 2013*) | | |
| Chemical compound, drug | Bodipy FL C5-ceramide | Thermo Fisher Scientific | Cat # D3521 | |
| Chemical compound, drug | Bodipy-Cholesterol | Avanti Polar Lipids | Cat # 810255 | |
| Chemical compound, drug | Bodipy FL C12-spinghomyelin | Thermo Fisher Scientific | Cat # D7711 | |
| Chemical compound, drug | Bodipy FL C5-spinghomyelin | Thermo Fisher Scientific | Cat # D3522 | |
| Chemical compound, drug | Bodipy FL succinimidyl ester (EverFluor FL) | Setareh Biotech | Cat # 7150 | |
| Chemical compound, drug | D-lactosyl-ß1–1' -D-erythro-sphingosine | Avanti Polar Lipids | Cat # 860542 | |
| Chemical compound, drug | D-lactosyl-ß—1,1' N-palmitoyl-D-erythro-sphingosine | Avanti Polar Lipids | Cat # 860576 | |
| Chemical compound, drug | Cholesterol | Sigma Aldrich | Cat # C3045 | |
| Chemical compound, drug | N-Palmitoyl-D-sphingomyelin | Sigma Aldrich | Cat# 91553 | |
| Chemical compound, drug | Sphingomyelinase (SMase) (from *Bacillus cereus*) | Sigma Aldrich | Cat# S7651 | |
| Chemical compound, drug | Myriocin (from *Mycelia sterilia*) | Sigma Aldrich | Cat# M1177 | |
| Chemical compound, drug | Paraformaldehyde 16% | Electron Microscopy Sciences | Cat # 15710 | |
| Chemical compound, drug | Paraformaldehyde | Thermo Fisher Scientific | Cat # PA0095 | |
| Chemical compound, drug | Glutaraldehyde 25% | Taab Laboratory Equipment Ltd | Cat # G011/2 | |
| Chemical compound, drug | 3-isobutyl-1-methylxanthine (IBMX) | Sigma Aldrich | Cat # I5879 | |
| Chemical compound, drug | Dexamethasone (Dex) | Sigma Aldrich | Cat # D4902 | |
| Chemical compound, drug | Insulin | Sigma Aldrich | Cat # I0516 | |
| Chemical compound, drug | rosiglitazone | Cayman Chemicals | 71740 | |
| Chemical compound, drug | Lipofectamine 2000 | Thermo Fisher Scientific | Cat # 11668019 | |
| Chemical compound, drug | *N,N*-Diisopropy lethylamine | Sigma Aldrich | Cat # D125806 | |
| Chemical compound, drug | *N,N*-Dimethyl formamide, anhydrous | Sigma Aldrich | Cat # 227056 | |
| Chemical compound, drug | Chloroform | VWR | Cat # VWRC22711.324 | |
| Chemical compound, drug | Methanol | Thermo Fisher Scientific | Cat # 15394528 | |

*Continued on next page*

Continued

| Reagent type (species) or resource | Designation | Source or reference | Identifiers | Additional information |
|---|---|---|---|---|
| Chemical compound, drug | Doxycycline hyclate | Sigma Aldrich | Cat # D9891 | |
| Chemical compound, drug | Hygromycin B | Thermo Fisher Scientific | Cat # 10687010 | |
| Chemical compound, drug | Blasticidin S HCl | Thermo Fisher Scientific | Cat # R210-01 | |
| Cell line (*Homo-sapiens*) | HeLa (cervix) | ATCC | Cat # CRM-CCL-2 RRID:CVCL_0030 | |
| Cell line (*M. musculus*) | 3T3-L1 (Pre-adipocytes) | ATCC | Cat # ATC-CL-173 RRID:CVCL_0123 | |
| Cell line (*Homo-sapiens*) | HeLa FlpIn T-Rex Caveolin1-mCherry | This study | | |
| Cell line (*Homo-sapiens*) | HeLa FlpIn T-Rex EHD2pTagBFP-P2A-Caveolin1-mCherry | This study | | |
| Transfected construct (human) | siRNA to EHD2 (Stealth) | Thermo Fisher Scientific | Cat#1299001 Assay ID HSS121265 | |
| Transfected construct (human) | Scrambled 353 siRNA (Stealth) | Thermo Fisher Scientific | Cat# 12935300 | |
| Transfected construct (human) | siRNA to Caveolin1 (Stealth) | Thermo Fisher Scientific | Cat #1299001 Assay ID HSS141467 | |
| Recombinant DNA reagent | Rab5-TagBFP | (*Francis et al., 2015*) | | under control of $P_{CMV\ IE}$ promotor in pTagBFP-N vector |
| Recombinant DNA reagent | EHD2-TagBFP | This study | | under control of $P_{CMV\ IE}$ promotor in pTagBFP-N vector |
| Recombinant DNA reagent | EHD2 I157Q-TagBFP | This study | | under control of $P_{CMV\ IE}$ promotor in pTagBFP-N vector |
| Recombinant DNA reagent | EHD2-mCherry | (*Hoernke et al., 2017*) | | under control of $P_{CMV\ IE}$ promotor in pmCherry-N1 vector |
| Recombinant DNA reagent | mCherry-EHD2 | This study | | under control of $P_{CMV\ IE}$ promotor in pmCherry-C1 vector |
| Recombinant DNA reagent | EHD2 I157Q-mCherry | This study | | under control of $P_{CMV\ IE}$ promotor in pmCherry-N1 vector |
| Recombinant DNA reagent | mCherry-EHD2 I157Q | This study | | under control of $P_{CMV\ IE}$ promotor in pmCherry-C1 vector |
| Recombinant DNA reagent | pCDNA/FRT/TO caveolin1-mCherry | This study | | under control of $P_{CMV\ IE}$ promotor |
| Recombinant DNA reagent | pCDNA/FRT/TO EHD2-pTagBFP-P2A-caveolin1-mCherry | This study | | under control of $P_{CMV\ IE}$ promotor |
| Software, algorithm | ImageJ/Fiji | Fiji (*Schindelin et al., 2012*) | RRID:SCR_002285 | http://fiji.sc/ |
| Software, algorithm | Imaris x64 9.1.2 | Bitplane | RRID:SCR_007370 | http://www.bitplane.com/imaris |
| Software, algorithm | Prism 5.0 | GraphPad | RRID:SCR_002798 | https://www.graphpad.com/scientific-software/prism/ |
| Software, algorithm | Photoshop CS6 | Adobe | RRID:SCR_014199 | https://www.adobe.com/se/products/photoshop.html |

*Continued on next page*

*Continued*

| Reagent type (species) or resource | Designation | Source or reference | Identifiers | Additional information |
|---|---|---|---|---|
| Software, algorithm | Illustrator CS6 | Adobe | RRID:SCR_010279 | https://www.adobe.com/se/products/photoshop.html |
| Software, algorithm | Maps 3.3 | FEI | | https://www.fei.com/software/maps/ |
| Software, algorithm | Zen interface 2.3 | Zeiss | RRID:SCR_013672 | https://www.zeiss.com/microscopy/int/products/microscope-software/zen.html |
| Software, algorithm | Nis Elements 4.3 | Nikon | RRID:SCR_014329 | https://www.nikoninstruments.com/en_EU/Products/Software |
| Other | Aluminum backed silica gel plates (median pore size 60 Å, fluorescent indicator 254 nm) | Fisher Scientific | Cat # 10517771 | |
| Other | Chromatography grade silica gel (0.035–0.070 mm, 60 Å) | Acros Organic | Cat # 240360050 | |
| Other | Formvar TEM grids | Taab Laboratory Equipment Ltd | Cat # F005 | |
| Other | CS-25R17 coverlips (TIRF) | Warner Instruments | Cat # 64–0735 | |
| Other | CS-25R15 coverlips | Warner Instruments | Cat # 64–0715 | |
| Other | Precision cover glasses thickness No. 1.5H | Paul Marienfeld GmbH and Co. KG | Cat # 0117520 | |
| Other | PD-10 columns | GE Healthcare | Cat # 17-0851-0 | |
| Other | MatTek dishes (35 mm dish, high tolerance 1.5) | MatTek Corporation | Cat # P35G-0.170–14 C | |
| Other | HCS LipidTOX Deep Red Neutral Lipid Stain | Thermo Fisher Scientific | Cat # H34477 | |
| Other | Dulbecco's Modified Eagle Medium (DMEM) | Thermo Fisher Scientific | Cat # 41966052 | |
| Other | Opti-MEM I Reduced Serum Medium | Thermo Fisher Scientific | Cat # 31985070 | |
| Other | Dulbecco's Modified Eagle Medium (DMEM), no phenol red | Thermo Fisher Scientific | Cat # 21063029 | |
| Other | Sodium pyruvate | Thermo Fisher Scientific | Cat # 11360039 | |
| Other | Dako Fluorescent Medium | Dako | Cat # S3023 | |
| Other | Alexa Fluor 647 C2 Maleimide | Thermo Fisher Scientific | Cat # A20347 | |
| Other | DAPI (4',6-Diamidino-2-Phenylindole, Dilactate) | Thermo Fisher Scientific | Cat # D3571 | |
| Other | Fetal bovine serum | Thermo Fisher Scientific | Cat # 16000044 | |

## Reagents

1,2-dioleoyl-sn-glycero-3-phosphoethanolamine (DOPE), 1,2-dioleoyl-3-trimethylammonium-propane (chloride salt) (DOTAP), TopFluor-cholesterol (Bodipy-Chol), TopFluor- phosphatidylethanolamine (Bodipy-PE), D-lactosyl-ß−1,1' N-palmitoyl-D-erythro-sphingosine [LacCer(d18:1/16:0)] and Lyso-

Lactosylceramide (Lyso-LacCer) were purchased from Avanti Polar Lipids Inc (Alabaster, AL, US). Bodipy FL C5-ganglioside GM1 (Bodipy-GM1), Bodipy FL C5-ceramide (Bodipy-Cer), Bodipy FL $C_{12}$-spinghomyelin (Bodipy-SM $C_{12}$), Bodipy FL $C_5$-spinghomyelin (Bodipy-SM $C_5$) were obtained from Thermo Fisher Scientific (Waltham, MA, US). BODIPY Fl-C5 NHS ester (4,4-Difluoro-5,7-dimethyl-4-bora-3a,4a-diaza-s-indacene-3-pentanoic acid, succinimidyl ester) was purchased from Setareh Biotech, LLC (Eugene, OR, US). Sphingomyelin (SM d18:1/16:0), cholesterol (Chol), d7-cholesterol, *N,N*-diisopropylethylamine, sphingomyelinase (SMase) from *Bacillus cereus*, myriocin from *Mycelia sterilia*, anhydrous dimethylforamide (DMF), chloroform ($CHCl_3$), and methanol (MeOH) were purchased from Sigma-Aldrich (St. Louis, MO, US). LC-MS grade formic acid was purchased from VWR Chemicals (Radnor, PA, US). LC-MS grade 2-propanol and acetonitrile were from Merck Millipore (Billerica, MA, US). Milli-Q water (Merck Millipore) was used. All reagents and chemicals were used without further purification.

## Bodipy-LacCer synthesis

Thin layer chromatography was performed on aluminum backed silica gel plates (median pore size 60 Å, fluorescent indicator 254 nm, Fisher Scientific, Hampton, NH, US), visualized by exposure to UV light (365 nm), and stained with acidic ethanolic vanillin solution. Flash chromatography was performed using chromatography grade silica gel (0.035–0.070 mm, 60 Å, Thermo Fisher Scientific). NMR spectra were recorded on a Bruker AVANCE (600 MHz) spectrometer. $^1$H Chemical shifts are reported in δ values relative to tetramethylsilane and referenced to the residual solvent peak ($CD_3OD$: $δ_H$ = 3.31 ppm, $δ_C$ = 49.00 ppm). Coupling constants are reported in Hz.

Lyso-LacCer (5 mg, 8 µM) was dissolved in DMF (200 µl) and *N,N*-diisopropylethylamine (2.1 µl, 12 µM, 1.5 eq.) was added. BODIPY Fl-C5 NHS ester (66 µl of a stock solution of 5 mg/100 µl DMF, 8 µM, 1.0 eq.) was added and the reaction was shielded from light and stirred for 14 h. The reaction mixture was concentrated and purified by column chromatography ($CHCl_3$, MeOH, $H_2O$, 70:15:2 – 65:25:2) to afford the product Bodipy-LacCer (6.5 mg, 88%, *Figure 1—figure supplement 2— source data 1*) as a red film (*Gretskaya and Bezuglov, 2013*).

Retention factor: $R_f$ = 0.46 ($CHCl_3$, MeOH, $H_2O$, 65:25:2).

NMR data: $^1$H-NMR ($CD_3OD$, 600 MHz) δ 7.41 (1H, s), 7.03 (1H, d, *J* = 4.1 Hz), 6.36 (1H, d, *J* = 4.0 Hz), 6.18 (1H, s), 5.67 (1H, dt, *J* = 15.3, 6.8 Hz), 5.44 (1H, dd, *J* = 15.3, 7.7 Hz), 4.34 (1H, d, *J* = 7.7 Hz), 4.29 (1H, d, *J* = 7.8 Hz), 4.17 (1H, dd, *J* = 10.1, 4.7 Hz), 4.07 (1H, t, *J* = 7.9 Hz), 3.99 (1H, ddd, *J* = 8.2, 4.6, 3.3 Hz), 3.89 (1H, dd, *J* = 12.1, 2.6 Hz), 3.84 (1H, dd, *J* = 12.2, 4.3 Hz), 3.81 (1H, d, *J* = 3.1 Hz), 3.78 (1H, dd, *J* = 11.4, 7.5 Hz), 3.70 (1H, dd, *J* = 11.5, 4.6 Hz), 3.61–3.59 (1H, m), 3.62–3.51 (3H, m), 3.52–3.49 (1H, m), 3.47 (1H, dd, *J* = 9.7, 3.3 Hz), 3.39 (1H, ddd, *J* = 9.3, 4.0, 2.7 Hz), 3.28 (1H, t, *J* = 8.5 Hz), 2.94 (2H, t, *J* = 7.3 Hz), 2.50 (3H, s), 2.28 (3H, s), 2.25 (2H, t, *J* = 7.0 Hz), 2.00–1.93 (2H, m), 1.79–1.66 (4H, m), 1.37–1.20 (31H, m, (11-$CH_2$)), 0.89 (3H, t, *J* = 7.0 Hz).

$^{13}$C-NMR ($CD_3OD$, 151 MHz) δ 175.7, 160.9, 160.2, 145.0, 136.1, 135.2, 134.9, 131.2, 130.0, 125.6, 120.9, 117.9, 105.1, 104.5, 80.6, 77.1, 76.5, 76.3, 74.8, 74.8, 73.0, 72.5, 70.3, 69.9, 62.5, 61.8, 54.8, 37.1, 33.4, 33.1, 30.8, 30.8, 30.8, 30.8, 30.8, 30.7, 30.5, 30.4, 30.3, 29.5, 29.4, 27.0, 23.7, 14.9, 14.5, 11.2.

## Cell lines and primary cultures

HeLa cells (ATCC-CRM-CCL-2, RRID:CVCL_0030) were cultured in Dulbecco's Modified Eagle Medium (DMEM, Thermo Fisher Scientific) supplemented with 10% (v/v) fetal bovine serum (FBS, Thermo Fisher Scientific) at 37°C, 5% $CO_2$. For generation of HeLa Flp-In T-REx Caveolin1-mCherry cells, the pcDNA/FRT/TO/Caveolin1-mCherry construct was generated by exchanging the EGFP-tag in the pcDNA/FRT/TO/Caveolin1-EGFP (*Mohan et al., 2015*) for a mCherry tag by restriction cloning using enzymes AgeI and NotI (Thermo Fisher Scientific). The HeLa Flp-In T-REx EHD2-BFP-P2A-Caveolin1-mCherry construct was generated by linearizing pcDNA/FRT/TO/Caveolin1-mCh with the restriction enzyme HindIII (Thermo Fisher Scientific). The DNA encoding EHD2-BFP and the P2A peptide was inserted by Gibson assembly using NEBuilder HiFi DNA assembly master mix (New England BioLabs, Ipswich, MA, USA). The Flp-In TRex HeLa cell lines were maintained in DMEM supplemented with 10% (v/v) FBS, 100 µg/ml hygromycin B (Thermo Fisher Scientific), and 5 µg/ml blasticidin S HCl (Thermo Fisher Scientific) for plasmid selection at 37°C, 5% $CO_2$. Expression at endogenous levels was induced by incubation with 0.5 ng/ml (Cav1-mCh) and 1.0 ng/ml (EHD2-BFP-

P2A-Cav1mCh) doxycycline hyclate (Dox, Sigma-Aldrich) for 16–24 h. All cell lines tested negative for mycoplasma.

3T3-L1 fibroblasts (ATC-CL-173, RRID:CVCL_0123) were maintained in DMEM supplemented with 10% (v/v) FBS and penicillin-streptomycin (10000 U/ml, 1:100, Thermo Fisher Scientific) at 37°C, 5% $CO_2$, and differentiated to adipocytes as previously described (Zebisch et al., 2012). Briefly, cells were either seeded directly into a six-well plate or on glass coverslips in a six-well plate at $6 \times 10^5$ cells/well (day −3 of differentiation). The cells reached confluency the following day and the medium was changed (day −2). After 48 h (day 0) the medium was exchanged for differentiation medium I [supplemented DMEM containing 0.5 mM 3-isobutyl-1-methylxanthine (IBMX, Sigma Aldrich), 0.25 µM dexamethasone (Dex, Sigma Aldrich), 1 µg/ml insulin (Sigma Aldrich) and 2 µM rosiglitazone (Cayman Chemical, Ann Arbor, MI, USA)]. Following incubation for 48 h, the medium was changed to differentiation medium II (supplemented DMEM containing 1 µg/ml insulin) (day 2). Experiments were performed on day 4 of differentiation.

## Fusogenic liposomes

Liposomes were prepared from a lipid mixture of DOPE, DOTAP, and either Bodipy-tagged lipid or unlabeled lipid at a ratio of 47.5:47.5:5. Lipid blends were in $MeOH:CHCl_3$ (1:3, v/v). Following the generation of a thin film using a stream of nitrogen gas, the vesicles were formed by addition of 20 mM HEPES (VWR, Stockholm, SE, pH 7.5, final lipid concentration 2.8 µmol/ml) and incubated for 1.5 h at room temperature. Glass beads were added to facilitate rehydration. The liposome dispersion was sonicated for 30 min (Transsonic T 310, Elma, Singen, DE). The hydrodynamic diameters (z-average) of the liposomes were measured using dynamic light scattering with a Malvern Zetasizer Nano-S (Malvern Instruments, Worcestershire, UK). Samples were diluted 1:100 in 20 mM HEPES (pH 7.5) and measured using a UV-transparent disposable cuvettes (Sarstedt, Nümbrecht, DE). The measurements were performed at 20°C. The Nano DTS Software 5.0 was used for acquisition and analysis of the data.

## Lipid quantification by LC-ESI-MS/MS

One day prior to experiment, cells were seeded in a six-well plate. Cells were left untreated or treated with 11.7 nmol/ml of the different fusogenic liposomes for 10 min at 37°C, 5% $CO_2$. The cells were washed three times with PBS and harvested in 500 µl MeOH by scraping. Counting revealed that approximately $4 \times 10^5$ cells were obtained per sample. For myriocin (2.5 µM) and SMase (0.01 U) treatment, cells were incubated for 24 h or 2 h, respectively. Extraction was performed using a mixer mill set to a frequency 30 Hz for 2 min, with one tungsten carbide bead added to each tube. Thereafter the samples were centrifuged at 4°C, 14000 RPM, for 10 min. A volume of 260 µl of the supernatant was transferred to micro vials and evaporated under $N_2$ (g) to dryness. The dried extracts were stored at −80°C until analysis. Calibration curves of Bodipy-labeled standards (Bodipy-SM $C_{12}$ and Bodipy-LacCer) as well as standards for endogenous LacCer and SM [LacCer(d18:1/16:0) and SM(d18:1/16:0)] were prepared prior to analysis. Stock solutions of each compound were prepared at a concentration of 500 ng/µl and stored at −20°C. A five-point calibration curve (0.025–0.4 ngl/µl) was prepared by serial dilutions [Bodipy-SM $C_{12}R^2$ = 0.9909; LacCer(d18:1/16:0) $R^2$ = 0.9945; Bodipy-LacCer $R^2$ = 0.9983; LacCer(d18:1/14:0) $R^2$ = 0.8742], except for endogenous SM(d18:1/16:0) where 0.025–10.0 ng/µl was used ($R^2$ = 0.9991). Samples and calibration curves were analyzed using a 1290 Infinitely system from Agilent Technologies (Waldbronn, Germany), consisting of a G4220A binary pump, G1316C thermostated column compartment, and G4226A autosampler with G1330B autosampler thermostat coupled to an Agilent 6490 triple quadrupole mass spectrometer equipped with a jet stream electrospray ion source operating in positive ion mode. Separation was achieved injecting 2 µl of each sample (resuspended in 20 µl of MeOH) onto a CSH $C_{18}2.1 \times 50$ mm, 1.7 µm column (Waters, Milford, MA, USA) held at 60°C in a column oven. The gradient eluents used were 60:40 acetonitrile:$H_2O$ (A) and 89:10.5:0.4 isopropanol:acetonitrile:water (B), both with 10 mM ammonium formate and 0.1% formic acid, with a flow rate of 500 µl/min. The initial conditions consisted of 15% B, and the following gradient was used with linear increments: 0–1.2 min (15–30% B), 1.2–1.5 (30–55% B), 1.5–4.0 (55% B), 4.0–4.8 (55–100% B), 4.8–6.8 (100% B), 7.1–8.0 (15% B). The MS parameters were optimized for each compound (Table 1). The fragmentor voltage was set at

**Table 1.** Retention times (RT), MRM-transition stages monitored (precursor ion and product ions) and collision energies of analyzed compounds.

| Compounds | MRM transition | | RT (min) | Collision energy (V) |
|---|---|---|---|---|
| | Precursor ion | Product ion | | |
| Bodipy-LacCer | 926.5 | 562.4 | 1.48 | 30 |
| LacCer(d18:1/16:0) | 862.6 | 520.5 | 2.84 | 20 |
| LacCer(d18:1/14:0) | 834.6 | 264.3 | 2.8 | 40 |
| SM(d18:1/16:0) | 703.6 | 184.1 | 2.9 | 30 |
| Bodipy-SM $C_{12}$ | 865.6 | 184.1 | 2.12 | 30 |

380 V, the cell accelerator voltage at 5 V and the collision energies from 20 to 30 V, nitrogen was used as collision gas.

Jet-stream gas temperature was at 150°C with a gas flow of 16 l/min. The sheath gas temperature was kept at 350°C with a gas flow of 11 l/min. The nebulizer pressure was set to 35 psi and the capillary voltage was set at 4 kV. The QqQ was run in Dynamic MRM Mode using a retention time delta of 0.8 min and 500 ms cycle scans. The data were quantified using MassHunter Quantitative Analysis (Agilent Technologies, Atlanta, GA, USA).

## Cholesterol quantification by GC-MS

One day prior to experiment cells were seeded in a six-well plate. Cells were left untreated or treated with 11.7 nmol/ml fusogenic liposomes for 10 min at 37°C, 5% $CO_2$. The cells were washed three times with PBS and harvested in 250 µl MeOH by scraping, and two wells were pooled to generate approximately $8 \times 10^5$ cells per 500 µl sample into Eppendorf tubes. Extraction was performed using a mixer mill set to a frequency 30 Hz for 2 min, with one tungsten carbide bead added to each tube. Obtained extracts were centrifuged at 4°C, 14000 RPM for 10 min. A volume of 300 µl of the collected supernatants was transferred to individual micro vials and the extracts were dried under $N_2$ (g) to dryness. Separate calibration curves were prepared for endogenous and d7-Chol. A six-point calibration curve spanning from 0 to 10 ng/µl was prepared for d7-Chol ($R^2$ = 0.9909). For endogenous Chol a six-point calibration curve spanning from 0 to 500 ng/µl was prepared ($R^2$ = 0.9969). Methyl stearate at a final concentration of 5 ng/µl was used as internal standard in both calibration curves. Derivatization was performed according to a previously published method (*Gullberg et al., 2004*). In detail, 10 µl of methoxyamine (15 µg/µl in pyridine) was added to the dry sample that was shaken vigorously for 10 min before it was left to react at room temperature. After 16 h, 10 µl of MSTFA was added, the sample was shaken and left to react for 1 h at room temperature. A volume of 10 µl of methyl stearate (15 ng/µl in heptane) was added before analysis. For d7-cholesterol quantification, 1 µl of the derivatized sample was injected by an Agilent 7693 autosampler, in splitless mode into an Agilent 7890A gas chromatograph equipped with a multimode inlet (MMI) and 10 m x 0.18 mm fused silica capillary column with a chemically bonded 0.18 µm DB 5 MS UI stationary phase (J and W Scientific). The injector temperature was 250°C. The carrier gas flow rate through the column was 1 ml min$^{-1}$, the column temperature was held at 60°C for 1 min, then increased by 60°C min$^{-1}$ to 300°C and held there for 2 min. The column effluent is introduced into the electron impact (EI) ion source of an Agilent 7000C QQQ mass spectrometer. The thermal AUX 2 (transfer line) and the ion source temperatures were 250°C and 230°C, respectively. Ions were generated by a 70 eV electron beam at an emission current of 35 µA and analyzed in dMRM-mode. The solvent delay was set to 3 min. For a list of MRM transitions see *Table 2*. For endogenous Chol analysis, the samples were reanalyzed in split mode (10:1) together with the Chol calibration curve. The data were quantified using MassHunter Quantitative Analysis (Agilent Technologies, Atlanta, GA, USA).

## Calculations of the number of PM lipids

The average PM area of fibroblast is around 3000 µm$^2$ (*Sheetz et al., 2006*), of which 23% is estimated to be occupied by proteins (*Dupuy and Engelman, 2008*), which translates into the average PM of a cell containing approximately $7 \times 10^9$ lipids (*Alberts et al., 2002*). Our data are in agreement with these reported values, with our measured values for SM(d18:1/16:0) being 40% of total

**Table 2.** MRM transitions for labeled and endogenous Chol.

| Compound | Comment | Precursor ion | MS1 resolution | Product ion | MS2 resolution | RT | RT delta min (total) |
|---|---|---|---|---|---|---|---|
| Methyl stearate | IS-std | 298 | Unit | 101.1 | Unit | 5.6 | 2 |
| Chol | Quant | 329 | Unit | 95 | Unit | 7.8 | 2 |
| Chol | Qual | 368 | Unit | 213 | Unit | 7.8 | 2 |
| d7-Chol | Quant | 336 | Unit | 95 | Unit | 7.8 | 2 |
| d7- Chol | Qual | 375 | Unit | 213 | Unit | 7.8 | 2 |

SM species (*Kjellberg et al., 2014*), and 21 mol% of PM lipids translating to $9.6 \times 10^9$ lipids in the PM.

## Assessment of lipid incorporation into the PM with live cell spinning disk microscopy

One day prior to the experiment, non-induced Cav1-mCh HeLa cells or 3T3-L1 adipocytes were seeded on glass coverslips (CS-25R17 or CS-25R15, Warner Instruments, Hamden, CT, US) in a six-well plate at $3 \times 10^5$ cells/well (37°C, 5% $CO_2$). Live cell experiments were conducted in phenol red-free DMEM (live cell medium, Thermo Fisher Scientific) supplemented with 10% FBS and 1 mM sodium pyruvate (Thermo Fisher Scientific) at 37°C in 5% $CO_2$. To follow the distribution of Bodipy throughout the PM, a POC mini two chamber (PeCon, Erbach, DE) was used that allowed addition of the fusogenic liposomes during data acquisition. Liposomes were added at a concentration of 7 nmol/ml and movies of confocal stacks were recorded every 30 s over a period of 5 min using a 63X lens and Zeiss Spinning Disk Confocal controlled by ZEN interface (RRID:SCR_013672) with an Axio Observer.Z1 inverted microscope, equipped with a CSU-X1A 5000 Spinning Disk Unit and an EMCCD camera iXon Ultra from ANDOR. For TIRF movies the same system was used but employing a 100X lens and an Axio Observer.Z1 inverted microscope equipped with an EMCCD camera iXonUltra from ANDOR. The increase in fluorescence intensity (FI) of the Bodipy signal was measured within circular ROIs, which were either evenly distributed over the PM seen in the confocal section or over the basal PM in the case of TIRF. The total FI was determined by calculating integrated density (area x FI), which was then background corrected. Ten ROIs per cell were analyzed using Zeiss Zen interface ($n$ = 3, two independent experiments). Based on lipids occupying 65 $\text{Å}^2$, which translates to $3.1 \times 10^6$ lipid molecules/$\mu m^2$ (*Dopico, 2007*), and that mean liposome diameter was 225 nm, corresponding to an area of 0.19 $\mu m^2$, we calculated that each liposome contained $0.6 \times 10^6$ lipids, of which 5% were Bodipy-labeled. To estimate the cell volume, the cell surface was segmented with the surface feature within the Imaris x64 9.1.2 RRID:SCR_007370 (Bitplane, Zurich, CH) using the mCherry fluorescence.

## Constructs, transfections, and cell treatments

pTagBFP-C (Evrogen, Moscow, RU) was used to generate the expression constructs of Rab5 and EHD2 wt or I157Q. The cavin1-GFP construct was a kind gift from Prof. Robert G. Parton. Cav1-mCh HeLa cells were transfected with Lipofectamine 2000 (Thermo Fisher Scientific) using Opti-MEM I reduced serum medium (Thermo Fisher Scientific) for transient protein expression. For EHD2 and Cav1 depletion, Cav1-mCh HeLa cells were transfected with either stealth siRNA, specific against human EHD2 or human Cav1, or scrambled control (all from Thermo Fisher Scientific) using Lipofectamine 2000 and Opti-MEM according to manufacturer's instructions unless otherwise stated. Cells were transfected twice over a period of 72 h before the experiment. Protein levels were analyzed by SDS-PAGE and immunoblotting using rabbit anti-EHD2, RRID:AB_2833022 (*Morén et al., 2012*) and rabbit anti-Cav1 antibodies, RRID:AB_303405 (Abcam, Cambridge, UK). Mouse anti-clathrin heavy chain, RRID:AB_397865 (clone 23, BD Transduction Laboratories, San Jose, CA, US) was used as loading control. Cells were treated with 2.5 μM myriocin in complete medium 24 h prior to harvesting. SMase was added to cells to generate a final concentration of 0.01 units in complete medium 2 h prior to harvesting or live cell imaging.

## Analysis of caveolae dynamics

To track caveolae dynamics, induced Cav1-mCh HeLa cells were treated with fusogenic liposomes (7 nmol/ml) and 5 min TIRF movies were recorded with an acquisition time of 3 s. Imaris software was used for tracking analysis of Cav1-mCh positive structures (no distinction between single caveolae and Rosetta's are made), which were segmented as spots and structures with a diameter of 0.4 µm and with an intensity quality of 2% were selected (*Figure 2—figure supplement 3*) as previously described (*Mohan et al., 2015*). The applied algorithm was based on Brownian motion with max distance travelled of 0.8 µm and a max gap size of 4. Experiments where cavin1-GFP or EHD2 (wt and mutant) was either transiently expressed or depleted were performed and analyzed the same way. Colocalization of cavin1-GFP or EHD2 (wt and mutant) to Cav1-mCh was quantified with Imaris software. Within a ROI, spots were created in one channel (e.g., red channel) and the second channel (e. g., blue channel) was masked. The masked spots show only colocalized red and blue spots and the percentage was correlated to the original channel. Analysis of the dynamic behavior of caveolae positive for or lacking EHD2-BFP was performed using double Flp-In EHD2-BFP Cav1-mCh HeLa cells. The tracking was done as described above and the data from the tracks of Cav1-mCh spots lacking EHD2-BFP were collected and removed from the data of Cav1-mCh spots positive for EHD2-BFP. Statistical analysis was performed on track duration(s) and track mean speed (µm/s) data and data areshown as fold change. All micrographs and acquired movies were prepared with Fiji, RRID:SCR_002285 (*Schindelin et al., 2012*) and Adobe Photoshop CS6, RRID:SCR_014199.

## Intracellular trafficking of lipids

Induced Cav1-mCh HeLa cells were seeded on glass coverslips (CS-25R15) in a six-well plate at $3 \times 10^5$ cells/well (37°C, 5% $CO_2$). On the following day, the cells were incubated with fusogenic liposomes (7 nmol/ml) for 15 min or 3 h. Rab5-BFP (*Francis et al., 2015*) was transiently expressed. To analyze the localization of lipids to lipid droplets (LDs), induced Cav1-mCh HeLa cells were treated with lipids for 15 min or 3 h, fixed and stained with HCS LipidTOX Deep Red Neutral Lipid Stain (1:200, Thermo Fisher Scientific). Confocal stacks were acquired on Zeiss Spinning Disk Confocal microscope. The colocalization of lipids to Rab5-positive structures or LD as well as localization of Cav1-mCh to Rab5-positive structures was analyzed as described above using a masking method in Imaris software. Micrographs were prepared with Fiji (RRID:SCR_002285) (*Schindelin et al., 2012*) and Adobe Photoshop CS6 (RRID:SCR_014199).

## Immunostaining

Induced Cav1-mCh HeLa cells were seeded on precision coverslips (No. 1.5H, Paul Marienfeld GmbH and Co. KG, Lauda-Königshofen, DE) in 24-well plates at $50 \times 10^3$ cells/well and incubated overnight (37°C, 5% $CO_2$). Following incubation with fusogenic liposomes (7 nmol/ml) for 1 h, the cells were washed three times with phosphate-buffered saline (PBS, pH 7.4). Cells were fixed with 4% PFA in PBS (Electron Microscopy Sciences, Hatfield, PA, US) and subsequent permeabilization and blocking was carried out simultaneously using PBS containing 5% goat serum and 0.05% saponin. Cells were then immunostained with rabbit anti-EHD2 (*Morén et al., 2012*) and rabbit anti-PTRF, RRID:AB_88224 (Abcam) followed by goat anti-rabbit IgG secondary antibody coupled to Alexa Fluor 647, RRID:AB_2535814 (Thermo Fisher Scientific) as previously described (*Lundmark et al., 2008*). Confocal images were acquired using the Zeiss Spinning Disk Confocal microscope (63X lens). Pearson colocalization coefficients were obtained using Imaris software applying the Coloc feature with automatic thresholding. All Pearson coefficients were derived from two independent experiments for the EHD2 stain. Analysis of the colocalization of cavin1 and Cav1-mCh was repeated once. Data from at least 30 images were analyzed with images containing two to three cells on average. For caveolae expression levels, induced (0.5 ng/ml Dox) and non-induced Cav1-mCh HeLa cells were seeded and fixed as described above. Cells were immunostained with rabbit anti-Caveolin1 (Abcam) followed by goat anti-rabbit IgG secondary antibody coupled to Alexa Fluor 488 (Thermo Fisher Scientific). Confocal images were acquired using the Zeiss Spinning Disk Confocal microscope (63X lens). Caveolae spots were counted using the Imaris software with the same preferences as described above. Micrographs were prepared using Fiji (*Schindelin et al., 2012*) and Adobe Photoshop CS6.

## FRAP experiments

Induced Cav1-mCh HeLa cells were seeded on glass coverslips (CS-25R15) in a six-well plate at $3 \times 10^5$ cells/well and incubated overnight (37°C, 5% CO$_2$). Cells were treated with 7 nmol/ml of Bodipy-labeled liposomes for 10 min followed by two washes with live cell media before imaging using TIRF using a Zeiss Axio Observer.Z1 inverted microscope. Three reference images were recorded before a ROI was photobleached for 1000 ms using maximal laser intensity (488 nm or 561 nm). The fluorescent recovery images were taken every 3 s for 5 min. For the lipid incorporation experiment, a region within the PM with homogeneous fluorescence was chosen. FRAP of the EHD2 mutants was performed the same way. For the Bodipy-LacCer, Bodipy-Chol, and Bodipy-SM C$_{12}$ accumulated in caveolae, FRAP was performed between 15 and 60 min after lipid addition and regions with structures positive for Cav1-mCh, EHD2-BFP, and Bodipy-lipid were selected and for the bulk PM a region lacking caveolae was selected. For FRAP experiments that quantified the recovery of Cav1-mCh, induced Cav1-mCh HeLa cells were untreated, depleted of EHD2 using siRNA, or incubated with Bodipy-LacCer liposomes. FRAP experiments were performed as described above using the Zeiss Spinning Disk Confocal microscope (63X lens). The signal recovery was monitored in focal plane close to the basal membrane. The intensities of the bleached regions were corrected for background signal and photobleaching of the cell. Data from at least 10 cells were collected per condition and mean FRAP recovery curves were plotted using Prism 5.0 (Graph-Pad, San Diego, CA, US; RRID:SCR_002798).

## Microinjection

Mouse EHD2 cysteine mutant construct (L303C,C96S, C138S, C356S) was expressed as N-terminal His$_6$-tag fusion proteins in *Escherichia coli* Rosetta (DE3) and purified (*Daumke et al., 2007*). Dithio-threitol was removed from the protein using PD-10 columns and the protein was labeled with Alexa Fluor 647 C2 Maleimide (Thermo Fisher Scientific) (*Hoernke et al., 2017*). The protein was diluted to a concentration of 0.5 mg/ml in 150 mM NaCl, 20 mM HEPES pH 7.5, and 1 mM MgCl$_2$. Cav1-mCh HeLa cells were transfected with siRNA and induced as described above. One day prior to the injection experiment, Cav1-mCh HeLa cells were seeded in MatTek dishes (35 mm dish, high toler-ance 1.5, MatTek Corporation, Ashland, MA, US) with a cell density of $3 \times 10^5$ cells/dish and induced with Dox. In the case of LacCer addition, the cells were treated with 7 nmol/ml of Bodipy-LacCer fusogenic liposomes for 10 min followed by two washes with live cell media before microin-jection. Microinjection was performed with Injectman NI2 coupled to the programmable microinjec-tor Femtojet (Eppendorf, Hamburg, DE). The protein was loaded in Femtotips II (Eppendorf) and injection was done with an injection pressure of 1.0 hPa, compensation pressure of 0.5 hPa, and injection time of 0.1 s. Live images were acquired on TIRF every 3 s for a total of 5 min using a Nikon Eclipse Ti-E inverted microscope with a 100X lens (Apochromat 1.49 Oil 0.13–0.20 DIC N2, Nikon) using NIS Elements (RRID:SCR_014329). Z-stacks of injected cells were captured using a 60X lens (Apochromat 1.4o Oil DIC, Nikon). Tracking of Cav1-mCh and colocalization analysis was done with Imaris as previously described.

## Correlative light electron microscopy

Cav1-mCh cells transiently expressing EHD2-GFP alone or treated with Bodipy-LacCer, Bodipy-Chol, or Bodipy-SM C$_{12}$ liposomes were fixed in 2% paraformaldehyde (PFA) and 0.2% glutaraldehyde (Taab Laboratory Equipment Ltd, Aldermaston, UK) in 0.1 M phosphate buffer (pH 7.4) for 1–2 h and then stored in 1% PFA at 4°C. For the grid preparation, the cells were scraped into the fixative solu-tion and washed three times with PBS (pH 7.4) and once with PBS containing 0.1% glycine (pH 7.4, Merck Millipore, Burlington, US). The cell pellet was embedded in 12% gelatin (Dr. Oetker, food grade) in 0.1 M phosphate buffer (pH 7.4). Blocks of around 1 mm$^2$ were cut and cryo-protected by overnight infiltration in 2.3 M sucrose (VWR) in 0.1 M phosphate buffer. Next, the blocks were plunge frozen in liquid nitrogen. The sample block was sectioned at −120°C to obtain 80 nm sec-tions. These were mounted in a drop of in 0.1 M phosphate buffer containing 1:1 of 2% methyl cellu-lose (Sigma-Aldrich) and 2.3 M sucrose on TEM grids with a carbon-coated Formvar film (Taab Laboratory Equipment Ltd). The grids were incubated with PBS (pH 7.4) at 37°C for 20 min and stained with DAPI (4′,6-Diamidino-2-Phenylindole, Dilactate, 1:1000 in PBS, pH 7.4, Thermo Fisher Scientific) before imaging on a Nikon Eclipse Ti-E inverted microscope with a 100X lens (Apochromat

1.49 Oil 0.13–0.20 DIC N2, Nikon). Low magnification images were taken at 20X for orientation on the grid and to aid the overlay of fluorescent microscopy images and the higher resolution images of TEM. Contrasting for TEM was done by embedding the grids in 1.8% methyl cellulose and 0.4% uranyl acetate (Polysciences, Inc, Hirschberg an der Bergstrasse, DE) solution prepared in water (pH 4) for 10 min in the dark. TEM was performed with a Talos 120C transmission electron microscope (FEI, Eindhoven, NL) operating at 120kV. Micrographs were acquired with a Ceta 16M CCD camera (FEI) using Maps 3.3 (FEI, Hillsboro, OR, US). The fluorescent images were overlaid atop TEM images of the same cells collected from the ultrathin section using Adobe Photoshop CS6.

## Electron microscopy

3T3-L1 cells were seeded on MatTek dishes (35 mm dish, high tolerance 1.5) and differentiated to adipocytes as described above. 3T3-L1 adipocytes were untreated or incubated with Bodipy-Chol liposomes for 45 min, washed with PBS, and fixed as follows. All chemical fixation steps were performed using a microwave (Biowave, TED PELLA, inc) unless stated and solutions were prepared and rinses were performed in 0.1M cacodylate buffer (Sigma-Aldrich) or water. Fixation of the cells was performed in 0.05% malachite green oxalate (Sigma-Aldrich) and 2.5% gluteraldehyde (Taab Laboratory Equipment Ltd, Aldermaston, UK) in cacodylate buffer. The samples were rinsed four times with cacodylate buffer and post-fixed with 0.8% $K_3Fe(CN)_6$ (Sigma-Aldrich) and 1% $OsO_4$ (Sigma-Aldrich) in cacodylate buffer and rinsed four times with cacodylate buffer. The samples were then stained with 1% aqueous tannic acid (Sigma-Aldrich). Following two rinses in cacodylate buffer and water, samples were stained with 1% aqueous uranyl acetate (Polysciences, Inc, Hirschberg an der Bergstrasse, DE). After four washes with water, samples were dehydrated in gradients of ethanol (25%, 50%, 75%, 90%, 95%, 100% and 100%) (VWR). The samples were infiltrated with graded series of hard grade spurr resin (Taab Laboratory Equipment Ltd, Aldermaston, UK) in ethanol (1:3, 1:1 and 3:1) and then left in 100% resin for 1 h at room temperature. The samples were later polymerized overnight at 60°, sectioned and imaged with a Talos 120C transmission electron microscope (FEI, Eindhoven, NL) operating at 120kV. To obtain quantitative data, segmentation of caveolae for measurement of bulb width and measurement of neck diameter for surface-connected caveolae was performed with 'icy' (*de Chaumont et al., 2012*). To extract bulb width and surface area, the 'active cells' plug-in was used with three points to make an elliptical contour that fitted individual caveolae. The neck diameter was obtained by drawing a ROI across the neck of surface-connected caveolae. The analysis was performed blinded and with randomized sections.

## Statistical analysis

Statistical analysis was carried out by two-tailed unpaired Student *t*-test for comparison with control samples using GraphPad Prism 5.0 software. All experiments were performed at least twice with data representing mean ± SEM unless otherwise stated.

# Acknowledgements

We acknowledge the Biochemical Imaging Center (BICU) and Umeå Core Facility Electron Microscopy (UCEM) at Umeå University and the National Microscopy Infrastructure, NMI (VR-RFI 2016–00968) for providing assistance. We especially thank Irene Martinez at BICU for assistance and expertise with image analysis and data visualization. We thank Mikkel Roland Holst for help with establishing the HeLa Flp-In T-REx Caveolin1-mCherry cells. This work was supported by the Swedish Cancer Society (CAN2014/746, CAN 2017/735, RL and MH), The Hagbergs Foundation (RL and MH), Kempe Foundation (LWKM) , and the Swedish Research Council (dnr 2017–04028, RL and EL).

# Additional information

## Funding

| Funder | Grant reference number | Author |
| --- | --- | --- |
| Vetenskapsrådet | dnr 2017-04028 | Richard Lundmark<br>Elin Larsson |

| Cancerfonden | CAN 2017/735 | Richard Lundmark<br>Madlen Hubert |
| Cancerfonden | CAN2014/746 | Madlen Hubert<br>Richard Lundmark |
| Hagberg Foundation | | Madlen Hubert<br>Richard Lundmark |
| Kempe Foundations | | Lindon WK Moodie |

The funders had no role in study design, data collection and interpretation, or the decision to submit the work for publication.

### Author contributions

Madlen Hubert, Conceptualization, Formal analysis, Validation, Investigation, Visualization, Methodology, Writing - original draft, Project administration, Writing - review and editing; Elin Larsson, Conceptualization, Formal analysis, Validation, Investigation, Visualization, Methodology, Writing - original draft, Writing - review and editing; Naga Venkata Gayathri Vegesna, Formal analysis, Investigation, Visualization, Methodology; Maria Ahnlund, Formal analysis, Methodology; Annika I Johansson, Formal analysis, Investigation, Methodology; Lindon WK Moodie, Resources, Formal analysis, Methodology; Richard Lundmark, Conceptualization, Resources, Formal analysis, Supervision, Funding acquisition, Validation, Investigation, Visualization, Methodology, Writing - original draft, Project administration, Writing - review and editing

### Author ORCIDs

Richard Lundmark (iD) https://orcid.org/0000-0001-9104-724X

### Decision letter and Author response

Decision letter https://doi.org/10.7554/eLife.55038.sa1
Author response https://doi.org/10.7554/eLife.55038.sa2

## Additional files

### Supplementary files

• Transparent reporting form

### Data availability

All data generated or analysed during this study are included in the manuscript, supporting files and source data files provided for each figure.

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
