## [Decision Letter]

**Acceptance summary:**

Caveolae are dynamic structures at the plasma membrane (PM) that can undergo scission and endocytosis. It has been a longstanding challenge to study the effect of PM lipids on caveolar structures. In this study, Hubert et al. present a solution through elegant use of fusogenic liposomes, which allows for acute perturbation of the PM lipid composition. The authors show that scission of caveolae from the cell surface is driven by cholesterol and glycosphingolipid accumulation in these structures.

**Decision letter after peer review:**

Thank you for submitting your article "Lipid accumulation promotes scission of caveolae" for consideration by *eLife*. Your article has been reviewed Suzanne Pfeffer as the Senior Editor, a Reviewing Editor, and three reviewers. The following individuals involved in review of your submission have agreed to reveal their identity: Shiro Suetsugu (Reviewer #1); Ivan R Nabi (Reviewer #2).

The reviewers have discussed the reviews with one another and the Reviewing Editor has drafted this decision to help you prepare a revised submission.

Summary:

Caveolae are dynamic structures at the plasma membrane (PM) than can undergo scission and endocytosis. It has been technically challenging to study in detail the effect of PM lipids on caveolar structures. Here, Hubert et al. present the original use of fusogenic liposomes to perturb the lipid composition of PMs and monitor changes in caveolae structures. All three reviewers were supportive and agreed that this work was an interesting contribution to this field. However, as listed below, they did raise several concerns that will require additional experiments and/or explanations in the text.

Essential revisions:

1) In the Abstract, the authors say that "cholesterol and glycosphingolipids specifically accumulate in caveolae, which decreases their neck diameter and drives their scission from the cell surface." The data supporting the scission seems to be the reduced track duration. However, the end of tracking does not necessarily mean the scission from cell surface and could be explained by disassembly into a flat membrane. Either rephrase or provide data that distinguishes between these possibilities.

2) In the Abstract, the authors say that "lipid-induced scission was counteracted by the ATPase EHD2. We propose that lipid accumulation in caveolae generates an intrinsically unstable domain prone to scission if not balanced by the restraining force of EHD2 at the neck." This requires further clarification. How does the EHD2 counteracted with the imbalance of lipid? The importance of the ATPase activity of EHD2 might also be better analyzed with the ATPase deficient mutant in addition to I157Q mutant.

3) The addition of lipids via fusogenic liposomes is the fundamental basis of the study and needs to be thoroughly characterized. Why is the lipid distribution by TIRF in Figure 1B homogenous but not in Figure 1F? How does this non-homogeneous distribution overlap with Cav1 spots? Does homogeneity of insertion vary for different lipids? Are lipid expression levels the same for different cells in the same population?

4) Caveolae dynamics are inferred from TIRF imaging of Cav1-mCherry. A key aspect of this approach is how caveolae are defined from Cav1 fluorescent spots, which needs to be more clearly defined in the text. The text states a set threshold was applied to define caveolae Cav1 spots but the Materials and methods section indicates that spots larger than 0.4 µm diameter were selected. Why was this size limit selected (caveolae are <100 nm and diffraction limit is ~200-250nm)? Was no intensity threshold applied? What is the intensity distribution of large vs small spots? To what extent is Cav1-mCherry expression similar between cells and how robust is the application of "caveolae" selection criteria between cells? From the movies, the number of bright large spots clearly varies between cells. If these are the ones the authors are considering to be caveolae then how to explain that some cells have so few caveolae? And how to exclude that these structures are rosettes, associated with EHD2 expression? How can the authors exclude the possibility that the changes in mobility they are seeing relate not to caveolae scission but to rosette fragmentation? Please include figures and videos showing the spots selected for study, along with superimposed tracks. A clear discussion of criteria used to assign Cav1 spots as individual caveolae should be included.

5) The fusogenic liposomes allow for the incorporation of lipid species specifically enriched in caveolae such as cholesterol, sphingomyelin, ceramides and glycosphingolipids. To prove that the regulation of caveolae stability is specific for caveolar lipids, it would be nice if the authors could show that caveolae dynamics are not perturbed after the addition of other phospholipids, found outside caveolae, such as PC, PA, PG. In parallel, the tracking experiments only focus on Cav1 dynamics upon liposome fusion (Figure 2 and Figure 3E-H). But it is not clear whether other non-caveolar PM protein (clathrin HC for instance) dynamics would be changed after addition of the selected lipids? These data would strongly support the specificity of the described phenomenon.

6) Figure 2A presents a schematic of how track durations are interpreted to reflect caveolae dynamics. The basis for these classifications is not clearly presented nor is it clear why they are required. Indeed, why even introduce the surface adjacent class which according to the quantification in Figure 2B is very much a minority of the tracks detected. The data in Figure 2B clearly shows that EDH2 siRNA reduces the number of tracks with slow track speed for essentially all track durations, clearly reflecting the increased scission of caveolae upon EDH2 knockdown. In light of authors concerns with respect to reliability of track duration, why not just use the simple classification of track speed used in 2B to analyze the effects of lipids on caveolae dynamics.

7) The effect of SM-C12 addition to EDH2 overexpressing cells is interesting. What is the effect of sphingomyelinase treatment on EDH2 OE cells?

8) Please include the data for control siRNA in Figure 3C and Figure 3F.

9) Some of the lipids exhibit less efficient incorporation (SM C5 and C12 and to a lesser extent Cer). Since all experiments are done with the same amount of liposomes, this could explain why tracking parameters (Figure 3D, F) are not different from control condition. Will adding 3 times more SM C5 perturb caveolae dynamics?

10) Is the lipid-induced caveolae scission purely driven by lipids through membrane curvature perturbations or is it mediated by membrane pinching proteins? The authors should test the involvement of dynamin2, which is known to associate with the neck of caveolae and favor their internalization. Less EHD2 is colocalized with Cav1-mCherry 1h after the fusion of liposomes containing GM1, LacCer or Chol but is Dynamin2 found more often with Cav1 positive structures under these conditions?

11) The diffusion of lipids on caveolae is an important part of this manuscript. Please include this point in the Abstract. Please highlight a comparison of FRAP at caveolae to that of the bulk PM is necessary for discussing the diffusion of lipids in caveolae and in bulk PM. In Figure 4D, the FRAP curves are of ROIs, but these ROIs are not specified.

12) In Figure 4C, it is not clear what the bleach zone is and in Figure 4D why is this analysis not extended to other lipids? Similar for the EM studies in Figure 5, why are these not performed for other lipids, in particular SM-C12?

13) Please clarify the difference between the upper and lower panels in Figure 5B and 5C. Also, clarify why is meant by different appearance between the upper and lower panels. The shape measurements were done only for the cholesterol-supplemented cells, this analysis should also be carried out for the LacCer-supplemented cells.

14) In Figure 5, if caveolae scission was enhanced, then some of them would go to endosomes. Is this the case? Some control experiments using, for instance, Bodipy-PE would be informative.

15) The results of Figure 6 show no obvious effect of Cav1 depletion on endosomal localization of Bodipy-lipids. So, after scission what is the fate of lipid filled internalized caveolae? Do they stay in the vicinity of PM and fuse back? What is the amount/fraction of lipids (LacCer) found in Rab5 positive structures after 15 minutes and 3 hours? Also, the possibility of caveolae flattening induced by lipid loading of the PM should be assayed. Figure 3B shows colocalization of Cav1-mCh with endogenous Cavin1 in maximum z-projection of confocal images. The authors should also monitor Cav1-Cavin colocalization by TIRF and compare the number of Cav1 positive structure/dot colocalizing with Cavin1, before and after addition of fusogenic liposomes, and see if this condition triggers caveolae disappearance/disassembly (not internalization).

16) Experiments where lipid composition is changed in the opposite direction would enhance the arguments of this paper. It would be useful to compare the effects of cholesterol depletion, by methyl-β-cyclodextrin treatment for example.

---

## [Author Response]

Essential revisions:1) In the Abstract, the authors say that "cholesterol and glycosphingolipids specifically accumulate in caveolae, which decreases their neck diameter and drives their scission from the cell surface." The data supporting the scission seems to be the reduced track duration. However, the end of tracking does not necessarily mean the scission from cell surface and could be explained by disassembly into a flat membrane. Either rephrase or provide data that distinguishes between these possibilities.

We agree with the reviewer that this is an essential conclusion of the manuscript and this is why we have performed so many experiments to test this hypothesis. We have now added more experiments and further improved the schematic Figure 2A and clarified the text regarding how the different states of caveolae and how they would be reflected in our data. We have also improved the description of how caveolae were assigned and tracked (see also point 4). The data that support our conclusion on scission is presented in Figure 2 and Figure 3 together with supplemental figures and text. In brief, we have analyzed the number of tracks before and after treatment and saw no significant difference (Figure 2—figure supplement 2C), which would be expected if caveolae were disassembled. We have now also clarified the text regarding this. We have analyzed the duration and speed of caveolae and find that the duration is decreased while speed is increased following incorporation of glycosphingolipids and Chol. This is consistent with scission of caveolae and we also control for this by the analysis of EHD2 KO cells in which scission of caveolae is increased (Moren et al., 2012, Stoeber et al., 2012, Matthaeus et al., 2020). The increased speed would however not be consistent with disassembled caveolae.

We go on to test our hypothesis in Figure 3. In Figure 3A, we analyze if the treatment results in altered colocalization with EHD2 (would indicate surface release) and cavin1 (would indicate disassembly). We find that colocalization with EHD2, but not cavin1 is affected by glycosphingolipids and Chol supporting scission. We have now complemented the experiments with new data as suggested under point 15. To further assess this, we have now performed new experiments where we have analyzed the colocalization of cavin1 and Cav1 in live cells using TIRF before and after treatment. These data show that the colocalization between cavin1 and Cav1 is not altered following lipid incorporation again supporting that the detected effect is due to scission (See also response to comment 15).

Furthermore, as stated in the manuscript, caveolae scission would result in more mobile caveolae inside cells as previously shown. In Figure 3C we show that indeed lipid addition induces more mobile caveolae as does EHD2 depletion. Disassembly or flattening of caveolae would not result in this. We can also suppress the lipid-induced effect by overexpressing EHD2, which again indicates that the lipid-effect induces scission. To control for that the EHD2 expression is not due to secondary effect due to long term expression of EHD2, we microinject EHD2 and observe the same effect. In addition, in Figure 3E, if lipid addition and EHD2 depletion both induce caveolae scission then there should not be an additive effect of these treatments and indeed this is what we detect in Figure 3E.

In summary, given these experiments and results, we conclude that the increased dynamics of caveolae following increased levels of glycosphingolipids and Chol are most likely via scission of caveolae and therefore we have kept this statement in the Abstract.

2) In the Abstract, the authors say that "lipid-induced scission was counteracted by the ATPase EHD2. We propose that lipid accumulation in caveolae generates an intrinsically unstable domain prone to scission if not balanced by the restraining force of EHD2 at the neck." This requires further clarification. How does the EHD2 counteracted with the imbalance of lipid? The importance of the ATPase activity of EHD2 might also be better analyzed with the ATPase deficient mutant in addition to I157Q mutant.

In the revised manuscript, we have now more clearly presented the previously described mechanistic cycle for how EHD2 assemble and stabilize the caveolae neck to prevent scission and restrain caveolae to the cell surface via assembly into ring-like oligomeres (Daumke et al., 2007, Hoernke et al., 2017). Here, we go on to show that the accumulation of specific lipids in caveolae promote scission and destabilize their surface association but that this can be restrained by EHD2 (overexpression or microinjection). Therefore, we propose that the components (both proteins and lipids) building up the caveolae bulb creates a highly curved intrinsically unstable domain prone to scission, but that EHD2 provides a restraining force at the caveolae neck preventing scission.

Regarding the role of ATP-hydrolysis, we have previously generated and characterized different mutants of EHD2 and their effects on binding and hydrolysis of ATP as well as caveolae (Daumke et al., 2007, Moren et al., 2012). The ATPase impaired mutant T94A cause overassembly of EHD2 and membrane tubulation (since the regulatory ATPase activity is impaired). This mutant tubulates the caveolae neck, and it is therefore difficult to draw direct conclusions for how ATP-hydrolysis in itself influence the ability of EHD2 to stabilize lipid-induced scission of caveolae based on this mutant. In this work, we aimed to use the I157Q mutant, where ATP-hydrolysis is uncoupled to ATP-hydrolysis, which therefore form oligomeres with a slow dissociation time but without overassembly. This allows us to study if oligomerization of EHD2 at the caveolae neck is sufficient to prevent lipid induce scission or weather the stabilization requires constant adaptation of EHD2 assembly/dissassembly via rounds of ATP-hydrolysis. We have made this clearer in the text.

3) The addition of lipids via fusogenic liposomes is the fundamental basis of the study and needs to be thoroughly characterized. Why is the lipid distribution by TIRF in Figure 1B homogenous but not in Figure 1F? How does this non-homogeneous distribution overlap with Cav1 spots? Does homogeneity of insertion vary for different lipids? Are lipid expression levels the same for different cells in the same population?

We agree that this is an important part of the study, and we have now added text in the Results section that addresses these questions and refers to our extensive characterization throughout the manuscript (see Figure 1B, 1F, Figure 2C, Figure 4A-D, Figure 2—figure supplement 2A, Figure 2—video 2, Figure 2—video 3, Figure 2—video 4, Figure 3—figure supplement 1A, Figure 4—figure supplement 2A-D) to better characterize the lipid fusion and distribution of the lipids in the beginning of the Results section.

Lipid incorporation was similar for all cells in the population as also observed in Figure 1B, Figure 2C and Figure 2—figure supplement 2A. The even distribution of lipids in the PM was observed for all lipid species (see data throughout manuscript). Occasionally, however, bright stable spots and some enrichment of lipids in cellular protrusions were observed independent of lipid species as was shown in Figure 1B. These structures did not colocalize with Cav-1. We have now changed Figure 1F in order not to cause confusion.

4) Caveolae dynamics are inferred from TIRF imaging of Cav1-mCherry. A key aspect of this approach is how caveolae are defined from Cav1 fluorescent spots, which needs to be more clearly defined in the text.

Thank you for bringing it to our attention that the spot analysis could be explained in more detail. This has now been clarified in the Materials and method section. We have also added a detailed figure that outlines the method in Imaris software for illustration of spot selection (Figure 2—figure supplement 3).

The text states a set threshold was applied to define caveolae Cav1 spots but the Materials and methods section indicates that spots larger than 0.4 µm diameter were selected. Why was this size limit selected (caveolae are <100 nm and diffraction limit is ~200-250nm)?

Several caveolae-mCh spots were empirically measured from TIRF images and spots with a diameter of 0.4 µm rendered the best spot analysis as compared to the movies (Figure 2—figure supplement 3). This has been clarified in the Materials and Method section and an according literature reference added.

Was no intensity threshold applied? What is the intensity distribution of large vs small spots?

Yes, an intensity threshold was applied that was based on intensity quality of the spots and we have added all relevant details to the Materials and method section. A detailed figure that outlines the method in Imaris software has been added for illustration of spot selection (Figure 2—figure supplement3).

To what extent is Cav1-mCherry expression similar between cells and how robust is the application of "caveolae" selection criteria between cells? From the movies, the number of bright large spots clearly varies between cells. If these are the ones the authors are considering to be caveolae then how to explain that some cells have so few caveolae?

We have now added a figure (Figure 1—figure supplement 4), which shows that the Dox induced Cav1-mCh expression is similar to the endogenous caveolae levels as compared to non-induced cells that had been fixed and stained against Caveolin1. Caveolae from fixed and live cells were counted using the Imaris software and similar numbers of caveolae was identified showing that the application is reliable.

And how to exclude that these structures are rosettes, associated with EHD2 expression? How can the authors exclude the possibility that the changes in mobility they are seeing relate not to caveolae scission but to rosette fragmentation?

We do see some spots that are larger and of greater intensity than others which could be rosettas or just caveolae close to each other. In the CLEM analysis of caveolae in Hela cells we do only rarely detect rosettas. However, using TIRF we don’t have the resolution to determine this. Occasionally, we have observed that one spot can break up into two spots although this is rare. We do not observe any increase in the number of caveolae spots following treatments which would be the case if many clusters or rosettas are fragmented. Furthermore, the observed increase in speed of caveolae following incorporation of Chol and sphingolipids is too fast to be explained by increased lateral diffusion of individual caveolae. Fragmentation of rosettas in terms of scission would be indistinguishable from scission of individual caveolae.

Please include figures and videos showing the spots selected for study, along with superimposed tracks. A clear discussion of criteria used to assign Cav1 spots as individual caveolae should be included.

A detailed figure that outlines the method in Imaris software has been added for illustration of spot selection (Figure 2—figure supplement 3).

5) The fusogenic liposomes allow for the incorporation of lipid species specifically enriched in caveolae such as cholesterol, sphingomyelin, ceramides and glycosphingolipids. To prove that the regulation of caveolae stability is specific for caveolar lipids, it would be nice if the authors could show that caveolae dynamics are not perturbed after the addition of other phospholipids, found outside caveolae, such as PC, PA, PG. In parallel, the tracking experiments only focus on Cav1 dynamics upon liposome fusion (Figure 2 and Figure 3E-H). But it is not clear whether other non-caveolar PM protein (clathrin HC for instance) dynamics would be changed after addition of the selected lipids? These data would strongly support the specificity of the described phenomenon.

We have now clarified in the text that we do not only analyze lipids known to be enriched in caveolae and that the precise lipidomics of caveolae is not known. The lipids in our study are carefully selected, where we use a glycerophospholipid PE (not caveolae enriched), glycosphingolipids LacCer and GM1 (proposed to be enriched), Ceramide (not enriched, but used to compare to glycosylated LacCer and GM1), sphingomyelin (proposed to be enriched) and Chol (proposed to be enriched). We agree that there are more lipids that could be interesting to study but that this would be out of the scope for this study.

We do not intend to claim that an altered lipid composition will specifically affect only caveolae dynamics. This could also affect for example clathrin coated pits (CCP), but, in order to be useful such studies would have to be put into context of CCP-formation and dynamics, which would require a complete design dedicated to such system. Such measurement would require setting up and controlling this system and performing all relevant controls to validate potential effects as we have now done for the caveolae system in this manuscript. Therefore, although interesting, we believe that this would need to be addressed in further studies.

6) Figure 2A presents a schematic of how track durations are interpreted to reflect caveolae dynamics. The basis for these classifications is not clearly presented nor is it clear why they are required. Indeed, why even introduce the surface adjacent class which according to the quantification in Figure 2B is very much a minority of the tracks detected. The data in Figure 2B clearly shows that EDH2 siRNA reduces the number of tracks with slow track speed for essentially all track durations, clearly reflecting the increased scission of caveolae upon EDH2 knockdown. In light of authors concerns with respect to reliability of track duration, why not just use the simple classification of track speed used in 2B to analyze the effects of lipids on caveolae dynamics.

We agree that the classification was not clearly described, and we thank the reviewer for pointing this out. We have now reworked the schematic figure (Figure 2A) and the section of the text describing the classification. We also agree that the track speed should be used to analyze the effects on caveolae dynamics along with duration and have now clarified this in the text.

7) The effect of SM-C12 addition to EDH2 overexpressing cells is interesting. What is the effect of sphingomyelinase treatment on EDH2 OE cells?

To address this, we have performed new experiments analyzing the effects of SMase on caveolae dynamics when EHD2 is overexpressed. The data is presented in the text and in Figure 3F and shows that EHD2 can partly restrain the decreased duration and increased speed of caveolae induced by reduced SM-levels.

8) Please include the data for control siRNA in Figure 3C and Figure 3F.

We have added the data for control siRNA in Figure 3C. We assume that the reviewer is referring to Figure 3E instead of F, and we would like to point out that the data for control siRNA is presented in Figure 3—figure supplement 2B.

9) Some of the lipids exhibit less efficient incorporation (SM C5 and C12 and to a lesser extent Cer). Since all experiments are done with the same amount of liposomes, this could explain why tracking parameters (Figure 3D, F) are not different from control condition. Will adding 3 times more SM C5 perturb caveolae dynamics?

We have now performed experiments where 3 times more liposomes containing SMC5 has been added to cells. The incorporation of this lipid was thereby increased (Figure 2D), but yet no effect on the dynamics of caveolae was detected (Figure 2—figure supplement 2D).

10) Is the lipid-induced caveolae scission purely driven by lipids through membrane curvature perturbations or is it mediated by membrane pinching proteins? The authors should test the involvement of dynamin2, which is known to associate with the neck of caveolae and favor their internalization. Less EHD2 is colocalized with Cav1-mCherry 1h after the fusion of liposomes containing GM1, LacCer or Chol but is Dynamin2 found more often with Cav1 positive structures under these conditions?

We completely agree that this is a very interesting question. However, it will require very far-reaching mechanistic studies to elucidate this in order not to jump the conclusions. We are performing an extensive study of the role of dynamin2 in caveolae scission in collaboration with several research groups. This work is still in progress, but it is clear that our data are not fully consistent with current literature on the role of dynamin2, which is very interesting. We aim to put together a comprehensive study of dynamin2 and caveolae and therefore the addition of individual data regarding dynamin2 in the current manuscript would risk to cause distort the overall conclusions on the role of dynamin2.

11) The diffusion of lipids on caveolae is an important part of this manuscript. Please include this point in the Abstract. Please highlight a comparison of FRAP at caveolae to that of the bulk PM is necessary for discussing the diffusion of lipids in caveolae and in bulk PM. In Figure 4D, the FRAP curves are of ROIs, but these ROIs are not specified.

Thanks, these are very good suggestions, and we have now specified the ROIs where recovery curves are measured and included the comparison of the recovery in caveolae and in the surrounding PM (Figure 4E and Figure 4—figure supplement 3). We agree that this is an important part of the manuscript and have added lipid diffusion and sequestering to the Abstract.

12) In Figure 4C, it is not clear what the bleach zone is and in Figure 4D why is this analysis not extended to other lipids? Similar for the EM studies in Figure 5, why are these not performed for other lipids, in particular SM-C12?

Thanks for pointing this out. We have now made a clearer presentation of the bleached zone and the ROIs where the recovery is analyzed. We have now also extended the analysis to include SM C_12_, which is also heavily enriched in caveolae in EHD2 overexpressing cells (Figure 4—figure supplement 3). See also response to point 11.

13) Please clarify the difference between the upper and lower panels in Figure 5B and 5C. Also, clarify why is meant by different appearance between the upper and lower panels. The shape measurements were done only for the cholesterol-supplemented cells, this analysis should also be carried out for the LacCer-supplemented cells.

We apologize for not clearly describing the difference between upper and lower panels. This has now been added to the figure legend. We have also further explained how we use these data to quantify either the caveolae neck or the caveolae surface area.

The adipocyte cells were used as a system, where we can address the effect of Chol in a quantitative way since these cells are physiologically important for Chol homeostasis, the caveolea are enriched in Chol (Örtegren et.al., 2004), and these cells have numerous caveolae at the cell surface. Adipocyte cells do not and contain detectable levels of LacCer (Örtegren et.al., 2004). Therefore, we have not included analysis of LacCer incorporation since interpretation of such data, alongside Chol data, would be difficult given that we in that case would introduce a lipid not normally present in these cells. We have now clarified this in the text.

14) In Figure 5, if caveolae scission was enhanced, then some of them would go to endosomes. Is this the case? Some control experiments using, for instance, Bodipy-PE would be informative.

In the CLEM analysis of HeLa cells, we did observe occasional internal red spots (Cav1-mCherry) where caveolae-like structures were fused to what appeared like endosomes. However, these were quite rare, preventing quantitative statements, and without further labelling the endosomal-like compartment could not be verified. We have therefore not included this in the manuscript. We have quantified the amount of caveolae that colocalize with Rab5 before and after lipid-treatment and we do not detect a significant change (Figure 6C). As mentioned in the text and in the response to comment 15, most of the scissioned caveolae appear to remain as surface adjacent caveolae following lipid-induced scission.

15) The results of Figure 6 show no obvious effect of Cav1 depletion on endosomal localization of Bodipy-lipids. So, after scission what is the fate of lipid filled internalized caveolae? Do they stay in the vicinity of PM and fuse back? What is the amount/fraction of lipids (LacCer) found in Rab5 positive structures after 15 minutes and 3 hours? Also, the possibility of caveolae flattening induced by lipid loading of the PM should be assayed. Figure 3B shows colocalization of Cav1-mCh with endogenous Cavin1 in maximum z-projection of confocal images. The authors should also monitor Cav1-Cavin colocalization by TIRF and compare the number of Cav1 positive structure/dot colocalizing with Cavin1, before and after addition of fusogenic liposomes, and see if this condition triggers caveolae disappearance/disassembly (not internalization).

We do not see an altered number of spots in cells where caveolae are dynamic due to lipid addition, which suggest that they fission off and stay in vicinity or refuse. We have now clarified this in the text. As suggested, we have now also added a quantification of the amount of caveolae that colocalize to Rab5-positive structures before and after lipid-treatment. The percentages are quite low and we do detect a significant change following lipid treatment.

Thanks for the nice suggestion to analyze effects on caveolae disassembly/flattening by monitoring colocalization of Cav1 and cavin1 by TIRF. We have now performed these experiments before and after lipid treatments and the data is presented in Figure 3—figure supplement 1. We find that there is no significant difference in the amount of colocalization following incorporation of any of the lipids.

16) Experiments where lipid composition is changed in the opposite direction would enhance the arguments of this paper. It would be useful to compare the effects of cholesterol depletion, by methyl-β-cyclodextrin treatment for example.

As mentioned in the manuscript, acute, but controlled decrease of the levels of specific lipids is technically challenging. We have been able to use SMase-treatmentto reduce the levels of SM (Figure 2G). We have also performed many experiments to optimize the use of MBCD (time and concentrations) to reduce the levels of Chol as suggested. However, treatment with MBCD results in an immediate and drastic shrinkage of the cells, whereby the tracking is obscured and impossible to quantify. Therefore, we are not confident to use this methodology to lower Chol levels and measure effects on caveolae dynamics.